# Instance-Conditioned GAN

**Arantxa Casanova**
Facebook AI Research
École Polytechnique de Montréal
Mila, Quebec AI Institute

**Marlène Careil**
Facebook AI Research
Télécom Paris

**Jakob Verbeek**
Facebook AI Research

**Michał Drożdżal**[*]
Facebook AI Research

**Adriana Romero-Soriano**[*]
Facebook AI Research
McGill University

## Abstract

Generative Adversarial Networks (GANs) can generate near photo realistic images in narrow domains such as human faces. Yet, modeling complex distributions of datasets such as ImageNet and COCO-Stuff remains challenging in unconditional settings. In this paper, we take inspiration from kernel density estimation techniques and introduce a non-parametric approach to modeling distributions of complex datasets. We partition the data manifold into a mixture of overlapping neighborhoods described by a datapoint and its nearest neighbors, and introduce a model, called instance-conditioned GAN (IC-GAN), which learns the distribution around each datapoint. Experimental results on ImageNet and COCO-Stuff show that IC-GAN significantly improves over unconditional models and unsupervised data partitioning baselines. Moreover, we show that IC-GAN can effortlessly transfer to datasets not seen during training by simply changing the conditioning instances, and still generate realistic images. Finally, we extend IC-GAN to the class-conditional case and show semantically controllable generation and competitive quantitative results on ImageNet; while improving over BigGAN on ImageNet-LT. Code and trained models to reproduce the reported results are available at https://github.com/facebookresearch/ic_gan.

## 1 Introduction

Generative Adversarial Networks (GANs) [18] have shown impressive results in unconditional image generation [27, 29]. Despite their success, GANs present optimization difficulties and can suffer from mode collapse, resulting in the generator not being able to obtain a good distribution coverage, and often producing poor quality and/or low diversity generated samples. Although many approaches attempt to mitigate this problem – *e.g.* [20, 32, 35, 38] –, complex data distributions such as the one in ImageNet [45] remain a challenge for unconditional GANs [33, 36]. Class-conditional GANs [5, 39, 40, 56] ease the task of learning the data distribution by conditioning on class labels, effectively partitioning the data. Although they provide higher quality samples than their unconditional counterparts, they require labelled data, which may be unavailable or costly to obtain.

Several recent approaches explore the use of unsupervised data partitioning to improve GANs [2, 14, 17, 23, 33, 42]. While these methods are promising and yield visually appealing samples, their quality is still far from those obtained with class-conditional GANs. These methods make use of relatively coarse and non-overlapping data partitions, which oftentimes contain data points from different types of objects or scenes. This diversity of data points may result in a manifold with low

---

[*]Equal contribution.

35th Conference on Neural Information Processing Systems (NeurIPS 2021).

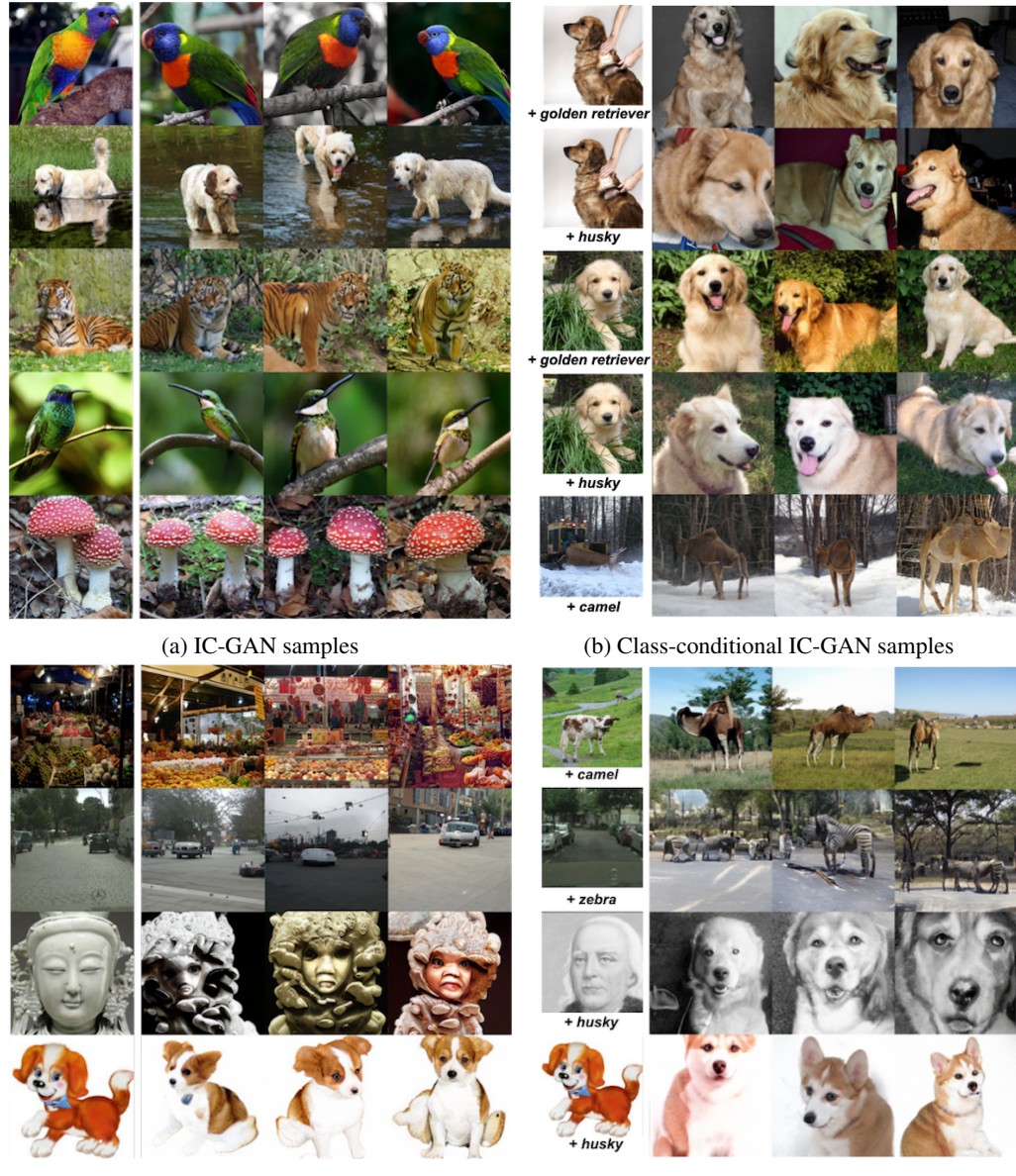

(a) IC-GAN samples

(b) Class-conditional IC-GAN samples

(c) IC-GAN transfer samples

(d) Class-conditional IC-GAN transfer samples

Figure 1: Samples from unlabeled (a) and class-conditional (b) IC-GAN trained on the $256 \times 256$ ImageNet dataset. For each subfigure, the first column represents instances used to condition the model and the next three columns depict model samples. For class-conditional generation in (b) we include samples conditioned on the same image but different labels. We highlight the generalization capacities of IC-GAN by applying the ImageNet-trained model to instances from other datasets in unlabeled (c) and class-conditional (d) scenarios. Panels (c) and (d) display samples conditioned on instances from the COCO-Stuff, Cityscapes, MetFaces, and PACS datasets (from top to bottom).

density regions, which degrades the quality of the generated samples [11]. Using finer partitions, however, tends to deteriorate results [33, 36, 42] because the clusters may contain too few data points for the generator and discriminator to properly model their data distribution.

In this work, we introduce a new approach, called instance-conditioned GAN (IC-GAN), which extends the GAN framework to model a mixture of local data densities. More precisely, IC-GAN learns to model the distribution of the neighborhood of a data point, also referred to as *instance*, by providing a representation of the instance as an additional input to both the generator and discriminator, and by using the neighbors of the instance as *real* samples for the discriminator. By choosing a sufficiently large neighborhood around the conditioning instance, we avoid the pitfall

of excessively partitioning the data into small clusters. Given the overlapping nature of these clusters, increasing the number of partitions does not come at the expense of having less samples in each of them. Moreover, unlike when conditioning on discrete cluster indices, conditioning on instance representations naturally leads the generator to produce similar samples for similar instances. Interestingly, once trained, our IC-GAN can be used to effortlessly transfer to other datasets not seen during training by simply swapping-out the conditioning instances at inference time.

IC-GAN bears similarities with kernel density estimation (KDE), a non-parametric density estimator in the form of a mixture of parametrized kernels modeling the density around each training data point – see *e.g.* [4]. Similar to KDE, IC-GAN can be seen as a mixture density estimator, where each component is obtained by conditioning on a training instance. Unlike KDE, however, we do not model the data likelihood explicitly, but take an adversarial approach in which we model the local density implicitly with a neural network that takes as input the conditioning instance as well as a noise vector. Therefore, the *kernel* in IC-GAN is no longer independent on the data point on which we condition, and instead of a kernel bandwidth parameter, we control the smoothness by choosing the neighborhood size of an instance from which we sample the *real* samples to be fed to the discriminator.

We validate our approach on two image generation tasks: (1) *unlabeled* image generation where there is no class information available, and (2) *class-conditional* image generation. For the unlabeled scenario, we report results on the ImageNet and COCO-Stuff datasets. We show that IC-GAN outperforms previous approaches in unlabeled image generation on both datasets. Additionally, we perform a series of transfer experiments and demonstrate that an IC-GAN trained on ImageNet achieves better generation quality and diversity when testing on COCO-Stuff than the same model trained on COCO-Stuff. In the class-conditional setting, we show that IC-GAN can generate images with controllable semantics – by adapting both class and instance–, while achieving competitive sample quality and diversity on the ImageNet dataset. Finally, we test IC-GAN in ImageNet-LT, a long-tail class distribution ablated version of ImageNet, highlighting the benefits of non-parametric density estimation in datasets with unbalanced classes. Figure 1 shows IC-GAN unlabeled ImageNet generations (a), IC-GAN class-conditional ImageNet generations (b), and IC-GAN transfer generations both in the unlabeled (c) and controllable class-conditional (d) setting.

## 2    Instance-conditioned GAN

The key idea of IC-GAN is to model the distribution of a complex dataset by leveraging fine-grained overlapping clusters in the data manifold, where each cluster is described by a datapoint $\mathbf{x}_i$ – referred to as *instance* – and its nearest neighbors set $\mathcal{A}_i$ in a feature space. Our objective is to model the underlying data distribution $p(\mathbf{x})$ as a mixture of *conditional distributions* $p(\mathbf{x}|\mathbf{h}_i)$ around each of $M$ instance feature vectors $\mathbf{h}_i$ in the dataset, such that $p(\mathbf{x}) \approx \frac{1}{M}\sum_i p(\mathbf{x}|\mathbf{h}_i)$.

More precisely, given an unlabeled dataset $\mathcal{D} = \{\mathbf{x}_i\}_{i=1}^{M}$ with $M$ data samples $\mathbf{x}_i$ and an embedding function $f$ parametrized by $\phi$, we start by extracting instance features $\mathbf{h}_i = f_\phi(\mathbf{x}_i) \ \forall \mathbf{x}_i \in \mathcal{D}$, where $f_\phi(\cdot)$ is learned in an unsupervised or self-supervised manner. We then define the set $\mathcal{A}_i$ of $k$ nearest neighbors for each data sample using the cosine similarity – as is common in nearest neighbor classifiers, *e.g.* [53, 54] – over the features $\mathbf{h}_i$. Figure 2a depicts a sample $\mathbf{x}_i$ and its nearest neighbors.

We are interested in implicitly modelling the conditional distributions $p(\mathbf{x}|\mathbf{h}_i)$ with a generator $G_{\theta_G}(\mathbf{z}, \mathbf{h}_i)$, implemented by a deep neural network with parameters $\theta_G$. The generator transforms samples from a unit Gaussian prior $\mathbf{z} \sim \mathcal{N}(0, I)$ into samples $\mathbf{x}$ from the conditional distribution $p(\mathbf{x}|\mathbf{h}_i)$, where $\mathbf{h}_i$ is the feature vector of an instance $\mathbf{x}_i$ sampled from the training data. In IC-GAN, we adopt an adversarial approach to train the generator $G_{\theta_G}$. Therefore, our generator is jointly trained with a discriminator $D_{\theta_D}(\mathbf{x}, \mathbf{h}_i)$ that discerns between real neighbors and generated neighbors of $\mathbf{h}_i$, as shown in Figure 2b. Note that for each $\mathbf{h}_i$, real neighbors are sampled uniformly from $\mathcal{A}_i$.

Both $G$ and $D$ engage in a two player min-max game where they try to find the Nash equilibrium for the following equation:

$$\min_G \max_D \mathbb{E}_{\mathbf{x}_i \sim p(\mathbf{x}), \mathbf{x}_n \sim \mathcal{U}(\mathcal{A}_i)}[\log D(\mathbf{x}_n, f_\phi(\mathbf{x}_i))] +$$
$$\mathbb{E}_{\mathbf{x}_i \sim p(\mathbf{x}), \mathbf{z} \sim p(\mathbf{z})}[\log(1 - D(G(\mathbf{z}, f_\phi(\mathbf{x}_i)), f_\phi(\mathbf{x}_i)))]. \tag{1}$$

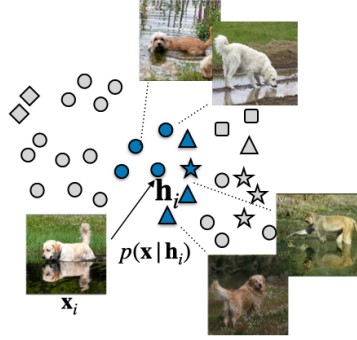
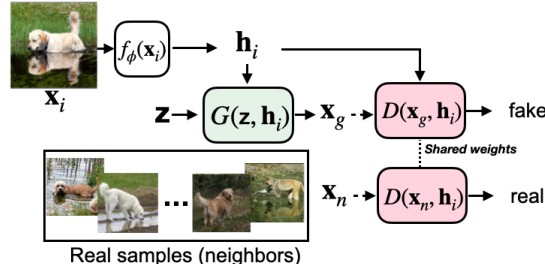

(a) Neighborhood $\mathcal{A}_i$ of instance $\mathbf{h}_i$       (b) Schematic illustration of the IC-GAN workflow

Figure 2: Overview of IC-GAN. (a) The goal of the generator is to generate realistic images similar to the neighbors of $\mathbf{h}_i$, defined in the embedding space using cosine similarity. Five out of seven neighbors are shown in the figure. Note that images in the same neighborhood may belong to different classes (depicted as different shapes). (b) Conditioned on instance features $\mathbf{h}_i$ and noise $\mathbf{z}$, the generator produces a synthetic sample $\mathbf{x}_g$. Generated samples and real samples (neighbors of $\mathbf{h}_i$) are fed to the discriminator, which is conditioned on the same $\mathbf{h}_i$.

Note that when training IC-GAN we use all available training datapoints to condition the model. At inference time, as in non-parametric density estimation methods such as KDE, the generator of IC-GAN also requires instance features, which may come from the training distribution or a different one.

**Extension to class-conditional generation.** We extend IC-GAN for class-conditional generation by additionally conditioning the generator and discriminator on a class label $\mathbf{y}$. More precisely, given a labeled dataset $\mathcal{D}_l = \{(\mathbf{x}_i, \mathbf{y}_i)\}_{i=1}^{M}$ with $M$ data sample pairs $(\mathbf{x}_i, \mathbf{y}_i)$ and an embedding function $f_\phi$, we extract instance features $\mathbf{h}_i = f_\phi(\mathbf{x}_i) \; \forall \mathbf{x}_i \in \mathcal{D}_l$, where $f_\phi(\cdot)$ is learned in an unsupervised, self-supervised, or supervised manner. We then define the set $\mathcal{A}_i$ of $k$ nearest neighbors for each data sample using the cosine similarity over the features $\mathbf{h}_i$, where neighbors may be from different classes. This results in neighborhoods, where the number of neighbors belonging to the same class as the instance $\mathbf{h}_i$ is often smaller than $k$. During training, real neighbors $\mathbf{x}_j$ and their respective labels $\mathbf{y}_j$ are sampled uniformly from $\mathcal{A}_i$ for each $\mathbf{h}_i$. In the class-conditional case, we model $p(\mathbf{x}|\mathbf{h}_i, \mathbf{y}_j)$ with a generator $G_{\theta_G}(\mathbf{z}, \mathbf{h}_i, \mathbf{y}_j)$ trained jointly with a discriminator $D_{\theta_D}(\mathbf{x}, \mathbf{h}_i, \mathbf{y}_j)$.

## 3 Experimental evaluation

We describe our experimental setup in Section 3.1, followed by results presented in the unlabeled setting in Section 3.2, dataset transfer in Section 3.3 and class-conditional generation in Section 3.4. We analyze the impact of the number of stored instances and neighborhood size in Section 3.5.

### 3.1 Experimental setup

**Datasets.** We evaluate our model in the unlabeled scenario on ImageNet [45] and COCO-Stuff [6]. The ImageNet dataset contains 1.2M and 50k images for training and evaluation, respectively. COCO-Stuff is a very diverse and complex dataset which contains multi-object images and has been widely used for complex scene generation. We use the train and evaluation splits of [8], and the (un)seen subsets of the evaluation images with only class combinations that have (not) been seen during training. These splits contain 76k, 2k, 675 and 1.3k images, respectively. For the class-conditional image generation, we use ImageNet as well as ImageNet-LT [34]. The latter is a long-tail variant of ImageNet that contains a subset of 115k samples, where the 1,000 classes have between 5 and 1,280 samples each. Moreover, we use some samples of four additional datasets to highlight the transfer abilities of IC-GAN: Cityscapes [10], MetFaces [28], PACS [31] and Sketches [15].

**Evaluation protocol.** We report Fréchet Inception Distance (FID) [22], Inception Score (IS) [47], and LPIPS [57]. LPIPS computes the distance between the AlexNet activations of two images generated with two different latent vectors and same conditioning. On ImageNet, we follow [5], and

compute FID over 50k generated images and the 50k real validation samples are used as reference. On COCO-Stuff and ImageNet-LT, we compute the FID for each of the splits using all images in the split as reference, and sample the same number images. Additionally, in ImageNet-LT we stratify the FID by grouping classes based on the number of train samples: more than 100 (many-shot FID), between 20 and 100 (med-shot FID), and less than 20 (few-shot FID). For the reference set, we split the validation images along these three groups of classes, and generate a matching number of samples per group. In order to compute all above-mentioned metrics, IC-GAN requires instance features for sampling. Unless stated otherwise, we store 1,000 training set instances by applying k-means clustering to the training set and selecting the features of the data point that is the closest to each one of the centroids. All quantitative metrics for IC-GAN are reported over five random seeds for the input noise when sampling from the model.

**Network architectures and hyperparameters.** As feature extractor $f_\phi$, we use a ResNet50 [21] trained in a self-supervised way with SwAV [7] for the unlabeled scenario; for the class-conditional IC-GAN, we use a ResNet50 trained for the classification task on either ImageNet or ImageNet-LT [26]. For ImageNet experiments, we use BigGAN [5] as a baseline architecture, given its superior image quality and ubiquitous use in conditional image generation. For IC-GAN, we replace the class embedding layers in the generator by a fully connected layer that takes the instance features as input and reduces its dimensionality from 2,048 to 512; the same approach is followed to adapt the discriminator. For COCO-Stuff, we additionally include the state-of-the-art unconditional StyleGAN2 architecture [29], as it has shown good generation quality and diversity in the lower data regime [28, 29]. We follow its class-conditional version [28] to extend it to IC-GAN by replacing the input class embedding by the instance features. Unless stated otherwise, we set the size of the neighborhoods to $k = 50$ for ImageNet and $k = 5$ for both COCO-Stuff and ImageNet-LT. See the supplementary material for details on the architecture and optimization hyperparameters.

## 3.2 Unlabeled setting

**ImageNet.** We start by comparing IC-GAN against previous work in Table 1. Note that unconditional BigGAN baseline is trained by setting all labels in the training set to zero, following [36, 42]. IC-GAN surpasses all previous approaches at both $64 \times 64$ and $128 \times 128$ resolutions in both FID and IS scores. At $256 \times 256$ resolution, IC-GAN outperforms the concurrent unconditional diffusion-based model of [12]; the only other result we are aware of in this setting. Additional results in terms of precision and recall can be found in Table 8 in the supplementary material.

As shown in Figure 1a, IC-GAN generates high quality images preserving most of the appearance of the conditioning instance. Note that generated images are not mere training memorizations; as shown in the supplementary material, generated images differ substantially from the nearest training samples.

**COCO-Stuff.** We proceed with the evaluation of IC-GAN on COCO-Stuff in Table 2. We also compare to state-of-the-art complex scene generation pipelines which rely on labeled bounding box annotations as conditioning – LostGANv2 [49] and OC-GAN [50]. Both of

Table 1: Results for ImageNet in unlabeled setting. For fair comparison with [42] at $64 \times 64$ resolution, we trained an unconditional BigGAN model and report the non-official FID and IS scores – computed with Pytorch rather than TensorFlow – indicated with *. [†]: increased parameters to match IC-GAN capacity. DA: 50% horizontal flips in (**d**) real and fake samples, and (**i**) conditioning instances. $ch \times$: Channel multiplier that affects network width as in BigGAN.

| Method | Res. | ↓FID | ↑IS |
|---|---|---|---|
| Self-sup. GAN [42] | 64 | 19.2* | 16.5* |
| Uncond. BigGAN[†] | 64 | 16.9* $\pm$ 0.0 | 14.6* $\pm$ 0.1 |
| **IC-GAN** | 64 | 10.4* $\pm$ 0.1 | 21.9* $\pm$ 0.1 |
| **IC-GAN** + DA (**d,i**) | 64 | **9.2*** $\pm$ 0.0 | **23.5*** $\pm$ 0.1 |
| MGAN [23] | 128 | 58.9 | 13.2 |
| PacGAN2 [32] | 128 | 57.5 | 13.5 |
| Logo-GAN-AE [46] | 128 | 50.9 | 14.4 |
| Self-cond. GAN [33] | 128 | 41.7 | 14.9 |
| Uncond. BigGAN [36] | 128 | 25.3 | 20.4 |
| SS-cluster GAN [36] | 128 | 22.0 | 23.5 |
| PGMGAN [2] | 128 | 21.7 | 23.3 |
| **IC-GAN** | 128 | 13.2 $\pm$ 0.0 | 45.5 $\pm$ 0.2 |
| **IC-GAN** + DA (**d,i**) | 128 | **11.7** $\pm$ 0.0 | **48.7** $\pm$ 0.1 |
| ADM [12] | 256 | 32.5 | 37.6 |
| **IC-GAN** ($ch \times 64$) | 256 | 17.0 $\pm$ 0.2 | 53.0 $\pm$ 0.4 |
| **IC-GAN** ($ch \times 64$) + DA (**d,i**) | 256 | 17.4 $\pm$ 0.1 | 53.5 $\pm$ 0.5 |
| **IC-GAN** ($ch \times 96$) + DA (**d**) | 256 | **15.6** $\pm$ 0.1 | **59.0** $\pm$ 0.4 |

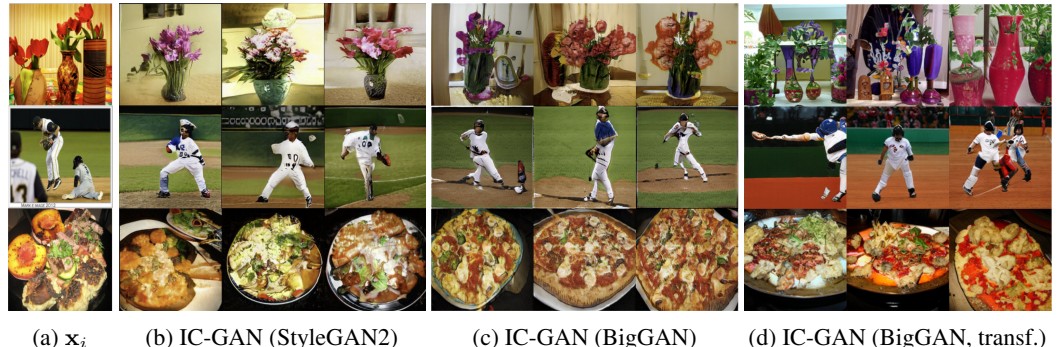

(a) $\mathbf{x}_i$      (b) IC-GAN (StyleGAN2)      (c) IC-GAN (BigGAN)      (d) IC-GAN (BigGAN, transf.)

Figure 3: Qualitative comparison for scene generation on $256 \times 256$ COCO-Stuff.

these approaches use tailored architectures for complex scene generation, which have at least twice the number of parameters of IC-GAN. Our IC-GAN matches or improves upon the unconditional version of the same backbone architecture in terms of FID in all cases, except for training FID with the StyleGAN2 backbone at $256 \times 256$ resolution. Overall, the StyleGAN2 backbone is superior to BigGAN on this dataset, and StyleGAN2-based IC-GAN achieves the state-of-the-art FID scores, even when compared to the bounding-box conditioned LostGANv2 and OC-GAN. IC-GAN exhibits notably higher LPIPS than LostGANv2 and OC-GAN, which could be explained by the fact that the latter only leverage one real sample per input conditioning during training; whereas IC-GAN uses multiple real neighboring samples per each instance, naturally favouring diversity in the generated images. As shown in figures 3b and 3c, IC-GAN generates high quality diverse images given the input instance. A qualitative comparison between LostGANv2, OC-GAN and IC-GAN can be found in Section E of the supplementary material.

Table 2: Quantitative results on COCO-Stuff. IC-GAN trained on ImageNet indicated as "transf". Some non-zero standard deviations are reported as 0.0 because of rounding.

| $128 \times 128$ | # prms. | train | $\downarrow$FID eval | eval seen | eval unseen | $\uparrow$LPIPS eval |
|---|---|---|---|---|---|---|
| LostGANv2 [49] | 41 M | $12.8 \pm 0.1$ | $40.7 \pm 0.3$ | $80.0 \pm 0.4$ | $55.2 \pm 0.5$ | $0.45 \pm 0.1$ |
| OC-GAN [50] | 170 M | — | $45.1 \pm 0.3$ | $85.8 \pm 0.5$ | $60.1 \pm 0.2$ | $0.13 \pm 0.1$ |
| Unconditional (BigGAN) | 18 M | $17.9 \pm 0.1$ | $46.9 \pm 0.5$ | $103.8 \pm 0.8$ | $60.9 \pm 0.7$ | $0.68 \pm 0.1$ |
| IC-GAN (BigGAN) | 22 M | $16.8 \pm 0.1$ | $44.9 \pm 0.5$ | $81.5 \pm 1.3$ | $60.5 \pm 0.5$ | $0.67 \pm 0.1$ |
| IC-GAN (BigGAN, transf.) | 77 M | $\mathbf{8.5} \pm 0.0$ | $\mathbf{35.6} \pm 0.2$ | $77.0 \pm 1.0$ | $\mathbf{48.9} \pm 0.2$ | $\mathbf{0.69} \pm 0.1$ |
| Unconditional (StyleGAN2) | 23 M | $8.8 \pm 0.1$ | $37.8 \pm 0.2$ | $92.1 \pm 1.0$ | $53.2 \pm 0.5$ | $0.68 \pm 0.1$ |
| IC-GAN (StyleGAN2) | 24 M | $8.9 \pm 0.0$ | $36.2 \pm 0.2$ | $\mathbf{74.3} \pm 0.8$ | $50.8 \pm 0.3$ | $0.67 \pm 0.1$ |
| $256 \times 256$ | | | | | | |
| LostGANv2 [49] | 46 M | $18.0 \pm 0.1$ | $47.6 \pm 0.4$ | $88.5 \pm 0.4$ | $62.0 \pm 0.6$ | $0.56 \pm 0.1$ |
| OC-GAN [50] | 190 M | — | $57.0 \pm 0.1$ | $98.7 \pm 1.2$ | $71.4 \pm 0.5$ | $0.21 \pm 0.1$ |
| Unconditional (BigGAN) | 21 M | $51.0 \pm 0.1$ | $81.6 \pm 0.5$ | $135.1 \pm 1.6$ | $95.8 \pm 1.1$ | $\mathbf{0.77} \pm 0.1$ |
| IC-GAN (BigGAN) | 26 M | $24.6 \pm 0.1$ | $53.1 \pm 0.4$ | $88.5 \pm 1.8$ | $69.1 \pm 0.6$ | $0.73 \pm 0.1$ |
| IC-GAN (BigGAN, transf.) | 90 M | $13.9 \pm 0.1$ | $\mathbf{40.9} \pm 0.3$ | $79.4 \pm 1.2$ | $\mathbf{55.6} \pm 0.6$ | $0.76 \pm 0.1$ |
| Unconditional (StyleGAN2) | 23 M | $\mathbf{7.1} \pm 0.0$ | $44.6 \pm 0.4$ | $98.1 \pm 1.7$ | $59.9 \pm 0.5$ | $0.76 \pm 0.1$ |
| IC-GAN (StyleGAN2) | 25 M | $9.6 \pm 0.0$ | $41.4 \pm 0.2$ | $\mathbf{76.7} \pm 0.6$ | $57.5 \pm 0.5$ | $0.74 \pm 0.1$ |

### 3.3 Off-the-shelf transfer to other datasets

In our first transfer experiment, we train IC-GAN with a BigGAN architecture on ImageNet, and use it to generate images from COCO-Stuff instances at test time. Quantitative results are reported as "IC-GAN (transf.)" in Table 2. In this setup, no COCO-Stuff images are used to train the model, thus, all splits contain unseen objects combinations. Perhaps surprisingly, IC-GAN trained on ImageNet outperforms the same model trained on COCO-Stuff for all splits: 8.5 *vs.* 16.8 train FID at 128 resolution. This raises the question of how close ImageNet and COCO-Stuff data distributions are. We compute the FID between real data train split of the two datasets at $128 \times 128$ resolution and obtain a score of 37.2. Hence, the remarkable transfer capabilities of IC-GAN are not explained by dataset similarity and may be attributed to the effectiveness of the ImageNet pre-trained feature

extractor and generator. When we replace the conditioning instances from COCO-Stuff with those of ImageNet, we obtain a train FID score of 43.5, underlining the important distribution shift that can be implemented by changing the conditioning instances.

Interestingly, the transferred IC-GAN also outperforms LostGANv2 and OC-GAN which condition on labeled bounding box annotations. Transferring the model from ImageNet boosts diversity w.r.t. the model trained on COCO-Stuff (see LPIPS in Table 2), which may be in part due to the larger $k = 50$ used for ImageNet training, compared to $k = 5$ when training on COCO-Stuff. Qualitative results of COCO-Stuff generations from the ImageNet pre-trained IC-GAN can be found in Figure 1c (top row) and Figure 3d. These generations suggest that IC-GAN is able to effectively leverage the large scale training on ImageNet to improve the quality and diversity of the COCO-Stuff scene generation, which contains significantly less data to train.

We further explore how the ImageNet trained IC-GAN transfers to conditioning on other datasets using Cityscapes, MetFaces, and PACS in Figure 1c. Generated images still preserve the semantics and style of the images for all datasets, although degrading their quality when compared to samples in Figure 1a, as the instances in these datasets –in particular MetFaces and PACS– are very different from the ImageNet ones. See Section F in the supplementary material for more discussion, additional evaluations, and more qualitative examples of dataset transfer.

## 3.4 Class-conditional setting

**ImageNet.** In Table 3, we show that the class-conditioned IC-GAN outperforms BigGAN in terms of both FID and IS across all resolutions except the FID at $128 \times 128$ resolution. It is worth mentioning that, unlike BigGAN, IC-GAN can control the semantics of the generated images by either fixing the instance features and swapping the class conditioning, or by fixing the class conditioning and swapping the instance features; see Figure 1b. As shown in the figure, generated images preserve semantics of both the class label and the instance, generating different dog breeds on similar backgrounds, or generating camels in the snow, an unseen scenario in ImageNet to the best of our knowledge. Moreover, in

Table 3: Class-conditional results on ImageNet. *: Trained using open source code. DA: 50% horizontal flips in (**d**) real and fake samples, and (**i**) conditioning instances. $ch\times$: Channel multiplier that affects network width. $^{\dagger}$: numbers from the original paper, as training diverged with the BigGAN opensourced code.

|  | Res. | ↓FID | ↑IS |
|---|---|---|---|
| BigGAN* [5] | 64 | $12.3 \pm 0.0$ | $27.0 \pm 0.2$ |
| BigGAN* [5] + DA (**d**) | 64 | $10.2 \pm 0.1$ | $30.1 \pm 0.1$ |
| IC-GAN | 64 | $8.5 \pm 0.0$ | $39.7 \pm 0.2$ |
| IC-GAN + DA(**d**, **i**) | 64 | $\mathbf{6.7} \pm 0.0$ | $\mathbf{45.9} \pm 0.3$ |
| BigGAN* [5] | 128 | $9.4 \pm 0.0$ | $98.7 \pm 1.1$ |
| BigGAN* [5] + DA(**d**) | 128 | $\mathbf{8.0} \pm 0.0$ | $107.2 \pm 0.9$ |
| IC-GAN | 128 | $10.6 \pm 0.1$ | $100.1 \pm 0.5$ |
| IC-GAN + DA(**d**, **i**) | 128 | $9.5 \pm 0.1$ | $\mathbf{108.6} \pm 0.7$ |
| BigGAN* [5] ($ch \times 64$) | 256 | $8.0 \pm 0.1$ | $139.1 \pm 0.3$ |
| BigGAN* [5] ($ch \times 64$) + DA(**d**) | 256 | $8.3 \pm 0.1$ | $125.0 \pm 1.1$ |
| IC-GAN ($ch \times 64$) | 256 | $8.3 \pm 0.1$ | $143.7 \pm 1.1$ |
| IC-GAN ($ch \times 64$) + DA(**d**, **i**) | 256 | $\mathbf{7.5} \pm 0.0$ | $152.6 \pm 1.1$ |
| BigGAN$^{\dagger}$ [5] ($ch \times 96$) | 256 | $8.1$ | $144.2$ |
| IC-GAN ($ch \times 96$) + DA(**d**) | 256 | $8.2 \pm 0.1$ | $\mathbf{173.8} \pm 0.9$ |

Figure 1d, we show the transfer capabilities of our class-conditional IC-GAN trained on ImageNet and conditioned on instances from other datasets, generating camels in the grass, zebras in the city, and husky dogs with the style of MetFaces and PACS instances. These controllable conditionings enable the generation of images that are not present or very rare in the ImageNet dataset, *e.g.* camels surrounded by snow or zebras in the city. Additional qualitative transfer results which either fix the class label and swap the instance features, or vice-versa, can be found in Section F of the supplementary material.

**ImageNet-LT.** Due to the class imbalance in ImageNet-LT, selecting a subset of instances with either k-means or uniform sampling can easily result in ignoring rare classes, and penalizing their generation. Therefore, for this dataset we use all available 115k training instances to sample from the model and compute the metrics. In Table 4 we compare to BigGAN, showing that IC-GAN is better in terms of FID and IS for modeling this long-tailed distribution. Note that the improvement is noticeable for each of the three groups of classes with different number of samples, see many/med/few column. In Section G of the supplementary material we present experiments when using class-balancing to train BigGAN, showing that it does not improve quality nor diversity of generated samples. We

Table 4: Class-conditional results on ImageNet-LT. *: Trained using open source code.

| | Res. | ↓train FID | ↑train IS | ↓val FID | many/med/few ↓val FID | ↑val IS |
|---|---|---|---|---|---|---|
| BigGAN* [5] | 64 | $27.6 \pm 0.1$ | $18.1 \pm 0.2$ | $28.1 \pm 0.1$ | $28.8 / 32.8 / 48.4 \pm 0.2$ | $16.0 \pm 0.1$ |
| IC-GAN | 64 | $\mathbf{23.2} \pm 0.1$ | $\mathbf{19.5} \pm 0.1$ | $\mathbf{23.4} \pm 0.1$ | $\mathbf{23.8 / 28.0 / 42.7} \pm 0.1$ | $\mathbf{17.6} \pm 0.1$ |
| BigGAN* [5] | 128 | $31.4 \pm 0.1$ | $30.6 \pm 0.1$ | $35.4 \pm 0.1$ | $34.0 / 43.5 / 64.4 \pm 0.2$ | $24.9 \pm 0.2$ |
| IC-GAN | 128 | $\mathbf{23.4} \pm 0.1$ | $\mathbf{39.6} \pm 0.2$ | $\mathbf{24.9} \pm 0.1$ | $\mathbf{24.3 / 31.4 / 53.6} \pm 0.3$ | $\mathbf{32.5} \pm 0.1$ |
| BigGAN* [5] | 256 | $27.8 \pm 0.0$ | $58.2 \pm 0.2$ | $31.4 \pm 0.1$ | $28.1 / 40.9 / 67.6 \pm 0.3$ | $44.7 \pm 0.2$ |
| IC-GAN | 256 | $\mathbf{21.7} \pm 0.1$ | $\mathbf{66.5} \pm 0.3$ | $\mathbf{23.4} \pm 0.1$ | $\mathbf{20.6 / 32.4 / 60.0} \pm 0.2$ | $\mathbf{51.7} \pm 0.1$ |

hypothesize that oversampling some classes may result in overfitting for the discriminator, leading to low quality image generations.

### 3.5 Selection of stored instances and neighborhood size

In this section, we empirically justify the k-means procedure to select the instances to sample from the model, consider the effect of the number of instances used to sample from the model, as well as the effect of the size $k$ of the neighborhoods $\mathcal{A}_i$ used during training. The impact of different choices for the instance embedding function $f_\phi(\mathbf{x})$ is evaluated in the supplementary material.

**Selecting instances to sample from the model.** In Figure 4 (left), we compare two instance selection methods in terms of FID: uniform sampling (Random) and k-means (Clustered), where we select the closest instance to each cluster centroid, using $k = 50$ neighbors during training (solid and dotted green lines). Random selection is consistently outperformed by k-means; selecting only 1,000 instances with k-means results in better FID than randomly selecting 5,000 instances. Moreover, storing more than 1,000 instances selected with k-means does not result in noticeable improvements in FID. Additionally, we computed FID metrics for the 1,000 ground truth images that are closest to the k-means cluster centers, obtaining $41.8 \pm 0.2$ FID, which is considerably higher than the $10.4 \pm 0.1$ FID we obtain with IC-GAN ($k = 50$) when using the same 1,000 cluster centers. This supports the idea that IC-GAN is generating data points that go beyond the stored instances, better recovering the data distribution.

We consider precision (P) and recall (R) [30] (using an InceptionV3 [51] as feature extractor and sampling 10,000 generated and real images) to disentangle the factors driving the improvement in FID, namely image quality and diversity (coverage) – see Figure 4 (right). We see that augmenting the number of stored instances results in slightly worse precision (image quality) but notably better recall (coverage). Intuitively, this suggests that by increasing the number of stored instances, we can better recover the data density at the expense of slightly degraded image quality in lower density regions of the manifold – see *e.g.* [11].

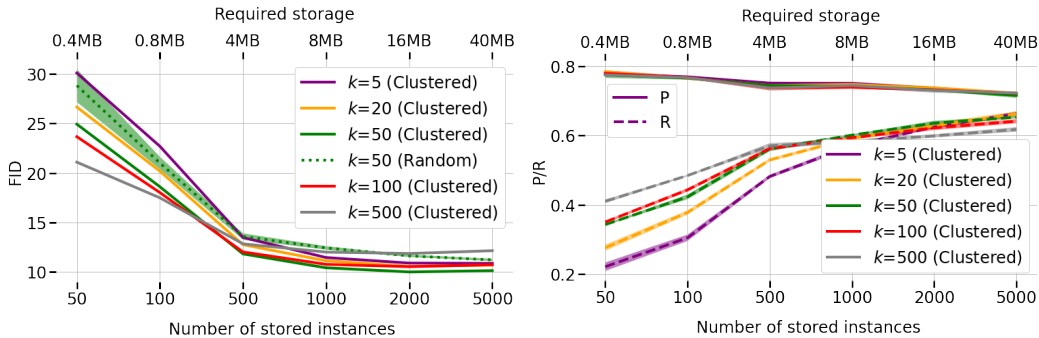

Figure 4: Impact on the number of stored instances used to evaluate IC-GAN and the size of the neighborhood $k$. Experiments performed on the $64 \times 64$ unlabeled ImageNet dataset.

**Neighborhood size.** In Figure 4 (both panels) we analyze the interplay between the neighborhood size and the number of instances used to recover the data distribution. For small numbers of stored instances, we observe that larger the neighborhoods lead to better (lower) FID scores (left-hand side of left panel). For recall, we also observe improvements for large neighborhoods when storing few instances (left-hand side of right panel), suggesting that larger neighborhoods are more effective in recovering the data distribution from few instances. This trend is reverted for large numbers of stored instances, where smaller values of $k$ are more effective. This supports the idea that the neighborhood size acts as a bandwidth parameter – similar to KDE –, that controls the smoothness of the implicitly learnt conditional distributions around instances. For example, $k = 500$ leads to smoother conditional distributions than $k = 5$, and as a result requires fewer stored instances to recover the data distribution. Moreover, as expected, we notice that the value of $k$ does not significantly affect precision (right panel). Overall, $k = 50$ offers a good compromise, exhibiting top performance across all metrics when using at least $500$ stored instances. We visualize the smoothness effect by means of a qualitative comparison across samples from different neighborhood sizes in Section K of the supplementary material. Using (very) small neighborhoods (*e.g.* of $k = 5$), results in lower diversity in the generated images.

# 4    Related work

**Data partitioning for GANs.**    Previous works have attempted to improve the image generation quality and diversity of GANs by partitioning the data manifold through clustering techniques [2, 19, 33, 36, 42, 46], or by leveraging mixture models in their design [14, 17, 23]. In particular, [36, 46] apply k-means on representations from a pre-trained feature extractor to cluster the data, and then use cluster indices to condition the generator network. Then, [19, 33] introduce an alternating two-stage approach where the first stage applies k-means to the discriminator feature space and the second stage trains a GAN conditioned on the cluster indices. Similarly, [42] proposes to train a clustering network, which outputs pseudolabels, in cooperation with the generator. Further, [2] trains a feature extractor with self-supervised pre-training tasks, and creates a k-nearest neighbor graph in the learned representation space to cluster connected points into the same sub-manifold. In this case, a different generator is then trained for each identified sub-manifold. By contrast, IC-GAN uses fine-grained overlapping data neighborhoods in tandem with conditioning on rich feature embeddings (instances) to learn a localized distribution around each data point.

**Mitigating mode collapse in GANs.** Works which attempt to mitigate mode collapse may also bear some similarities to ours. In [32], the discriminator takes into consideration multiple random samples from the same class to output a decision. In [35], a mixed batch of generated and real samples is fed to the discriminator with the goal of predicting the ratio of real samples in the batch. Other works use a mixture of generators [17, 23] and encourage each generator to focus on generating samples from a different mode. Similarly, in [14], the discriminator is pushed to form clusters in its representation space, where each cluster is represented by a Gaussian kernel. In turn, the generator tends to learn to generate samples covering all clusters, hence mitigating mode collapse. By contrast, we focus on discriminating between real and generated *neighbors* of an instance conditioning, by using a single generator network trained following the GAN formulation.

**Conditioning on feature vectors**. Very recent work [37] uses image self-supervised feature representations to condition a generative model whose objective is to produce a good input reconstruction; this requires storing the features of all training samples. In contrast, our objective is to learn a localized distribution (as captured by nearest neighboring images) around each conditioning instance, and we only need to save a very small subset of the dataset features to approximately recover the training distribution.

**Kernel density estimation and adversarial training.** Connections between adversarial training and nonparametric density estimation have been made in prior work [1]. However, to the best of our knowledge, no prior work models the dataset density in a nonparametric fashion with a localized distribution around each data point with a single conditional generation network.

**Complex scene generation.** Existing methods for complex scene generation, where natural looking scenes contain multiple objects, most often aim at controllability and rely on detailed conditionings such as a scene graphs [3, 25], bounding box layouts [48–50, 58], semantic segmentation masks [9, 43, 44, 52, 55] or more recently, freehand sketches [16]. All these methods leverage intricate pipelines to generate complex scenes and require labeled datasets. By contrast, our approach

relies on instance conditionings which control the global semantics of the generation process, and does not require any dataset labels. It is worth noting that complex scene generation is often characterized by unbalanced, strongly long tailed datasets. Long-tail class distributions negatively affect class-conditional GANs, as they struggle to generate visually appealing samples for classes in the tail [8]. However, to the best of our knowledge, no other previous work tackles this problem for GANs.

## 5 Discussion

**Contributions.** We presented instance-conditioned GAN (IC-GAN), which models dataset distributions in a non-parametric way by conditioning both generator and discriminator on instance features. We validated our approach on the unlabeled setting, showing consistent improvements over baselines on ImageNet and COCO-Stuff. Moreover, we showed through transfer experiments, where we condition the ImageNet-trained model on instances of other datasets, the ability of IC-GAN to produce compelling samples from different data distributions. Finally, we validated IC-GAN in the class-conditional setting, obtaining competitive results on ImageNet and surpassing the Big-GAN baseline on the challenging ImageNet-LT; and showed compelling controllable generations by swapping the class-conditioning given a fixed instance or the instance given a fixed conditioning.

**Limitations.** IC-GAN showed excellent image quality for labeled (class-conditional) and unlabeled image generation. However, as any machine learning tool, it has some limitations. First, as kernel density estimator approaches, IC-GAN requires storing training instances to use the model. Experimentally, we noticed that for complex datasets, such as ImageNet, using 1,000 instances is enough to approximately cover the dataset distribution. Second, the instance feature vectors used to condition the model are obtained with a pre-trained feature extractor (self-supervised in the unlabeled case) and depend on it. We speculate that this limitation might be mitigated if the feature extractor and the generator are trained jointly, and leave it as future work. Third, although, we highlighted excellent transfer potential of our approach to unseen datasets, we observed that, in the case of transfer to datasets that are *very* different from ImageNet, the quality of generated images degrades.

**Broader impacts.** IC-GAN brings with it several benefits such as excellent image quality in labeled (class-conditional) and unlabeled image generation tasks, and the transfer potential to unseen datasets, enabling the use of our model on a variety of datasets without the need of fine-tuning or re-training. Moreover, in the case of class-conditional image generation, IC-GAN enables controllable generation of content by adapting either the style – by changing the instance – or the semantics – by altering the class –. Thus, we expect that our model can positively affect the workflow for creative content generators. That being said, with improving image quality in generative modeling, there is some potential for misuse. A common example are *deepfakes*, where a generative model is used to manipulate images or videos well enough that humans cannot distinguish real from fake, with the intent to misinform. We believe, however, that open research on generative image models also contributes to better understand such synthetic content, and to detect it where it is undesirable. Recently, the community has also started to undertake explicit efforts towards detecting manipulated content by organizing challenges such as the Deepfake Detection Challenge [13].

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
