# Instance-Conditioned GAN: Supplementary Material

We provide additional material to support the main paper. We credit the used assets by citing their web links and licenses in Section A, and continue by describing the experimental setup and used hyperparameters in Section B. We compute Precision and Recall metrics on ImageNet in Section C, and we further compare BigGAN and StyleGAN2 backbones for IC-GAN on ImageNet in Section D. We provide additional qualitative results for both IC-GAN on ImageNet in Section E and IC-GAN off-the-shelf transfer results on other datasets in Section F. Moreover, we provide results when training BigGAN with class balancing on ImageNet-LT in Section G. Finally, we show further impact studies such as the choice of feature extractor (Section H), the number of conditionings used during training (Section I), matching storage requirements for unconditional counterparts of BigGAN and StyleGAN2 and IC-GAN (Section J) and the qualitative impact of neighborhood size $k$ for ImageNet, as well as quantitative results for ImageNet-LT and COCO-Stuff (Section K).

## A    Assets and licensing information

In Tables 5 and 6, we provide the links to the used datasets, repositories and their licenses. We use Faiss [24] for a fast computation of nearest neighbors and k-means algorithm leveraging GPUs, DiffAugment [59] for additional data augmentation when training BigGAN, and the pre-trained SwAV [7] and ResNet50 models on ImageNet-LT [26] to extract instance features.

Table 5: Links to the assets used in the paper.

| Name | GitHub link |
|---|---|
| ImageNet [45] | https://www.image-net.org |
| ImageNet-LT [34] | https://github.com/zhmiao/OpenLongTailRecognition-OLTR |
| COCO-Stuff [6] | https://cocodataset.org/ |
| Cityscapes [10] | https://www.cityscapes-dataset.com/ |
| MetFaces [28] | https://github.com/NVlabs/metfaces-dataset |
| PACS [31] | https://domaingeneralization.github.io/ |
| Sketches [15] | http://cybertron.cg.tu-berlin.de/eitz/projects/classifysketch/ |
| BigGAN [5] | https://github.com/ajbrock/BigGAN-PyTorch |
| StyleGAN2 [28] | https://github.com/NVlabs/stylegan2-ada-pytorch |
| Faiss [24] | https://github.com/facebookresearch/faiss |
| DiffAugment [59] | https://github.com/mit-han-lab/data-efficient-gans |
| PRDC [41] | https://github.com/clovaai/generative-evaluation-prdc |
| SwAV [7] | https://github.com/facebookresearch/swav |
| Pre-trained ResNet50 [26] | https://github.com/facebookresearch/classifier-balancing |

Table 6: Assets licensing information.

| Name | License |
|---|---|
| ImageNet [45] and ImageNet-LT [34] | Terms of access: https://www.image-net.org/download.php |
| COCO-Stuff [6] | https://www.flickr.com/creativecommons |
| Cityscapes [10] | https://www.cityscapes-dataset.com/license |
| MetFaces [28] | Creative Commons BY-NC 2.0 |
| PACS [31] | Unknown |
| Sketches [15] | Creative Commons Attribution 4.0 International |
| BigGAN [5] | MIT |
| StyleGAN2 [28] | NVIDIA Source Code |
| Faiss [24] | MIT |
| DiffAugment [59] | BSD 2-Clause "Simplified" |
| PRDC [41] | MIT |
| swAV [7] | Attribution-NonCommercial 4.0 International |
| Pre-trained ResNet50 [26] | BSD |

## B    Experimental setup and hyperparameters

We divide the experimental section into architecture modifications in Subsection B.1 and training and hyperparameter details in Subsection B.2.

## B.1  Architecture modifications for IC-GAN.

In our IC-GAN experiments, we leveraged BigGAN and StyleGAN2 backbones, and extended their architectures to handle the introduced instance conditionings.

When using BigGAN as a base architecture, IC-GAN replaces the class embedding layers in both generator and discriminator by fully connected layers. The fully connected layer in the generator has an input size of $2,048$ (corresponding to the feature extractor $f_\theta$'s dimensionality) and an output size $o_{dim}$ that can be adjusted. For all our experiments, we used $o_{dim} = 512$ – selected out of $\{256, 512, 1,024, 2,048\}$. The fully connected layer in the discriminator has a variable output size $n_{dim}$ to match the dimensionality of the intermediate unconditional discriminator feature vector – following the practice in BigGAN [5]. For the class-conditional IC-GAN, we use both the class embedding layers as well as the fully connected layers associated with the instance conditioning. In particular, we concatenate class embeddings (of dimensionality $c_{dim} = 128$) and instance embeddings (with dimensionality $o_{dim} = 512$). To avoid the rapid growth of parameters when using both class and instance embeddings, we use $n_{dim}/2$ as the output dimensionality for each of the embeddings in the discriminator, so that the resulting concatenation has a dimensionality of $n_{dim}$.

When using StyleGAN2 as a base architecture, we modify the class-conditional architecture of [28]. In particular, we replace the class embeddings layers with a fully connected layer of output dimensionality $512$ in the generator. The fully connected layer substituting the class embedding in the discriminator is of variable size. In this case, the instance features are concatenated with the noise vector at the input of the StyleGAN2's mapping network, creating a *style vector* for the generator. However, when it comes to the discriminator, the mapping network is only fed with the extracted instance features to obtain a modulating vector that is multiplied by the internal discriminator representation at each block.

All instance feature vectors $\mathbf{h}_i$ are normalized with $\ell_2$ norm before computing the neighborhoods and when used as conditioning for the GAN.

## B.2  Training details and hyperparameters

All models were trained while monitoring the training FID, and training was stopped according to either one of the following criteria: (1) early stopping when FID did not improve for $50$ epochs – or the equivalent number of iterations depending on the batch size –, or (2) when the training FID diverged. For BigGAN, we mainly explored the hyperparameter space around previously known and successful configurations [5, 42]. Concretely, we focused on finding the following best hyperparameters for each dataset and resolution: the batch size ($BS$), model capacity controlled by channel multipliers ($CH$), number of discriminator updates versus generator updates ($D_{updates}$), discriminator learning rate ($D_{lr}$) and generator learning rate ($G_{lr}$), while keeping all other parameters unchanged [5]. For StyleGAN, we also performed a hyperparameter search around previously known successful settings [28]. More precisely, we searched for the optimal $D_{lr}$ and $G_{lr}$ and R1 regularization weight $\gamma$ and used default values for the other hyperparameters.

**ImageNet.**  When using the BigGAN backbone, in the $64 \times 64$ resolution, we followed the experimental setup of [42], where: $BS = 256$, $CH = 64$, $D_{lr} = G_{lr} = 1\mathrm{e}{-}4$ and found that, although the unconditional BigGAN baseline achieves better metrics with $D_{updates} = 2$, IC-GAN and BigGAN do so with $D_{updates} = 1$. Note that we explored additional configurations such as increasing $BS$ or $CH$ but did not observe any improvement upon the aforementioned setup. In both the $128 \times 128$ and $256 \times 256$ resolutions, BigGAN hyperparameters were borrowed from [5]. For IC-GAN, we explored $D_{lr}, G_{lr} \in \{4\mathrm{e}{-}4, 2\mathrm{e}{-}4, 1\mathrm{e}{-}4, 4\mathrm{e}{-}5, 2\mathrm{e}{-}5, 1\mathrm{e}{-}5\}$ and $D_{updates} \in \{1, 2\}$. For $128 \times 128$, we used $BS = 2,048$, $CH = 96$ (as in [5]), $D_{lr} = 1\mathrm{e}{-}4$, $G_{lr} = 4\mathrm{e}{-}5$ and $D_{updates} = 1$. For $256 \times 256$, we set $BS = 2,048$ and $CH = 64$ (half capacity, therefore faster training) for both BigGAN and IC-GAN, and used $D_{lr} = G_{lr} = 1\mathrm{e}{-}4$ with $D_{updates} = 2$ for IC-GAN. When using the StyleGAN2 architecture both as a baseline and as a backbone, we explored $BS \in \{32, 64, 128, 256, 512, 1,024\}$, $D_{lr}, G_{lr} \in \{1\mathrm{e}{-}2, 7\mathrm{e}{-}3, 5\mathrm{e}{-}3, 2.5\mathrm{e}{-}3, 1\mathrm{e}{-}4, 5\mathrm{e}{-}4\}$ and $\gamma \in \{2\mathrm{e}{-}1, 1\mathrm{e}{-}2, 5\mathrm{e}{-}2, 1\mathrm{e}{-}1, 2\mathrm{e}{-}1, 5\mathrm{e}{-}1, 1, 2, 10\}$ and selected $BS = 64$ and $D_{lr} = G_{lr} = 2.5\mathrm{e}{-}3$ and $\gamma = 5\mathrm{e}{-}2$ for all resolutions.

**COCO-Stuff.**  When using BigGAN architecture, we explored $BS \in \{128, 256, 512, 2,048\}$ and $CH \in \{32, 48, 64\}$ and found $BS = 256$ and $CH = 48$ to be the best choice. We

searched for $D_{lr}, G_{lr} \in \{1e{-}3, 4e{-}4, 1e{-}4, 4e{-}5, 1e{-}5\}$ and $D_{updates} \in \{1, 2\}$. For both unconditional BigGAN and IC-GAN, we chose $D_{lr} = 4e{-}4$ and $G_{lr} = 1e{-}4$ in $128 \times 128$ and $D_{lr} = G_{lr} = 1e{-}4$ in $256 \times 256$. For both resolutions, unconditional BigGAN uses $D_{updates} = 2$ and IC-GAN, $D_{updates} = 1$. When using StyleGAN2 architecture, we tried several learning rates $D_{lr}, G_{lr} \in \{1e{-}3, 1.5e{-}3, 2e{-}3, 2.5e{-}3, 3e{-}3\}$ in combination with $\gamma \in \{2e{-}1, 1e{-}2, 5e{-}2, 1e{-}1, 2e{-}1, 5e{-}1, 1, 2, 10\}$. For the unconditional StyleGAN2 and IC-GAN trained at resolution $128 \times 128$, we chose $D_{lr} = G_{lr} = 2.5e{-}3$ with $\gamma = 5e{-}2$. At resolution $256 \times 256$, we found that $D_{lr} = G_{lr} = 3e{-}3$ with $\gamma = 0.5$ were optimal for IC-GAN while we obtained $D_{lr} = G_{lr} = 2e{-}3$ with $\gamma = 2e{-}1$ for the unconditional StyleGAN.

**ImageNet-LT.** We explored $BS \in \{128, 256, 512, 1,024, 2,048\}$ and $CH \in \{48, 64, 96\}$ and found $BS = 128$ and $CH = 64$ to be the best configuration. We explored $D_{lr}, G_{lr} \in \{1e{-}3, 4e{-}4, 1e{-}4, 4e{-}5, 1e{-}5\}$ and $D_{updates} \in \{1, 2\}$. In $64 \times 64$, we used $D_{lr} = 1e{-}3$, $G_{lr} = 1e{-}5$ and $D_{updates} = 1$ for both BigGAN and IC-GAN setup. In $128 \times 128$ and $256 \times 256$, we used $D_{lr} = G_{lr} = 1e{-}4$ and $D_{updates} = 2$.

**Data augmentation.** We use horizontal flips to augment the real data fed to the discriminator in all experiments, unless stated otherwise. For COCO-Stuff and ImageNet-LT, we found that using translations with the DiffAugment framework [59] improves FID scores, as the number of training samples is significantly smaller than ImageNet (5% and 10% the size of ImageNet, respectively). However, we did not see any improvement in ImageNet dataset and therefore we do not use DiffAugment in our ImageNet experiments. For ImageNet and COCO-Stuff, we augment the conditioning instance features $\mathbf{h}_i$ by horizontally flipping all data samples $\mathbf{x}_i$ and obtaining a new $\mathbf{h}_i$ from the flipped image, unless stated otherwise in the tables. This effectively doubles the number of conditionings available at training time, which have the same sample neighborhood $\mathcal{A}_i$ as their non-flipped versions. We tried applying this augmentation technique to ImageNet-LT but found that it degraded the overall metrics, possibly due to the different feature extractor used in these experiments. We hypothesize that the benefits of this technique are dependent on the usage of horizontal flips during the training stage of the feature extractor. As seen in Table 7, using data augmentation in the conditioning instance features slightly improves the results for IC-GAN both when coupled with BigGAN and StyleGAN2 backbones in COCO-Stuff.

**Compute resources.** We used NVIDIA V100 32GB GPUs to train our models. Given that we used different batch sizes for different experiments, we adapted the resources to each dataset configuration. In particular, ImageNet $64 \times 64$ models were trained using 1 GPU, whereas ImageNet $128 \times 128$ and $256 \times 256$ models required 32 GPUs. ImageNet-LT $64 \times 64$, $128 \times 128$ and $256 \times 256$ used 1, 2 and 8 GPUs each, respectively. Finally, COCO-Stuff $128 \times 128$ and $256 \times 256$ required 4 and 16 GPUs, respectively, when using the BigGAN backbone, but required 2 and 4 GPUs when leveraging StyleGAN2.

## C   Additional metrics: Precision and Recall

As additional measures of visual quality and diversity, we compute Precision (P) and Recall (R) [30] in Table 8. Results are provided on the ImageNet dataset, following the experimental setup proposed in [41]. By inspecting the results, we conclude that IC-GAN obtains better Recall (and therefore more diversity) than all the baselines in both the unlabeled and labeled settings, when selecting 10,000 random instances from the training set. Moreover, when selecting 1,000 instances with k-means, which is the standard experimental setup we used across the paper, we obtain higher Precision (as a measure of visual quality) than the other baselines in the unlabeled setting. In the labeled setting, the Precision is also higher than the one of BigGAN for 64x64 while being lower for $128 \times 128$ and $256 \times 256$.

## D   Comparison between StyleGAN2 and BigGAN backbones on ImageNet

We present additional experiments with IC-GAN using the StyleGAN2 backbone in ImageNet in Table 9, comparing them to StyleGAN2 across all resolutions. IC-GAN with a StyleGAN2 backbone obtains better FID and IS than StyleGAN2 across all resolutions, further supporting that IC-GAN does not depend on a specific backbone, as already shown in the COCO-Stuff dataset in

Table 7: Comparison between IC-GAN with and without data augmentation using the COCO-Stuff dataset. [†]: 50% chance of horizontally flipping data samples $\mathbf{x}_i$ to later obtain $\mathbf{h}_i$. The backbone for each IC-GAN is indicated with the number of parameters between parentheses. To compute FID in the training split, we use a subset of $1,000$ training instance features (selected with k-means) as conditionings.

| | Backbone (M) | ↓FID | | | |
| --- | --- | --- | --- | --- | --- |
| | | train | eval | eval seen | eval unseen |
| 128x128 | | | | | |
| IC-GAN | BigGAN (22) | $18.0 \pm 0.1$ | $45.5 \pm 0.7$ | $85.0 \pm 1.1$ | $60.6 \pm 0.9$ |
| IC-GAN [†] | BigGAN (22) | $\mathbf{16.8} \pm 0.1$ | $\mathbf{44.9} \pm 0.5$ | $\mathbf{81.5} \pm 1.3$ | $\mathbf{60.5} \pm 0.5$ |
| IC-GAN | StyleGAN2 (24) | $8.9 \pm 0.0$ | $36.2 \pm 0.2$ | $74.3 \pm 0.8$ | $50.8 \pm 0.3$ |
| IC-GAN [†] | StyleGAN2 (24) | $\mathbf{8.7} \pm 0.0$ | $\mathbf{35.8} \pm 0.1$ | $74.0 \pm 0.7$ | $\mathbf{50.5} \pm 0.6$ |
| 256x256 | | | | | |
| IC-GAN | BigGAN (26) | $25.6 \pm 0.1$ | $53.2 \pm 0.3$ | $91.1 \pm 3.3$ | $\mathbf{68.3} \pm 0.9$ |
| IC-GAN [†] | BigGAN (26) | $\mathbf{24.6} \pm 0.1$ | $\mathbf{53.1} \pm 0.4$ | $\mathbf{88.5} \pm 1.8$ | $69.1 \pm 0.6$ |
| IC-GAN | StyleGAN2 (24.5) | $10.1 \pm 0.0$ | $41.8 \pm 0.3$ | $78.5 \pm 0.9$ | $57.8 \pm 0.6$ |
| IC-GAN [†] | StyleGAN2 (24.5) | $\mathbf{9.6} \pm 0.0$ | $\mathbf{41.4} \pm 0.2$ | $\mathbf{76.7} \pm 0.6$ | $\mathbf{57.5} \pm 0.5$ |

Table 2. StyleGAN2, despite being designed for unconditional generation, is outperformed by the unconditional counterpart of BigGAN, that uses a single label for the entire dataset, in ImageNet. We suspect that there might be some biases introduced in the architecture at design time, as BigGAN was proposed for ImageNet and StyleGAN2 was tested on datasets with human faces, cars, and dogs, generally with presumably lower complexity and less number of data points than ImageNet. This intuition is further supported by StyleGAN2 improving over the BigGAN backbone in the COCO-Stuff experiments in Table 2, as this dataset is much smaller than ImageNet and contains a lot of images where people are depicted. Interestingly, we qualitatively found that people and their faces are better generated with a StyleGAN2 backbone rather than the BigGAN one when trained on COCO-Stuff.

# E    Additional qualitative results for IC-GAN

**Unlabeled ImageNet.**    IC-GAN generates high quality and diverse images that generally preserve the semantics and style of the conditioning. Figure 5 shows three instances – a golden retriever in the water, a humming bird on a branch, and a landscape with a castle –, followed by their six closest nearest neighbors in the feature space of SwAV [7], a ResNet50 model trained with self-supervision. Note that, although all neighbors contain somewhat similar semantics to those of the instance, the class labels do not always match. For example, one of the nearest neighbors of a golden retriever depicts a monkey in the water. The generated images depicted in Figure 5 are obtained by conditioning IC-GAN with a BigGAN backbone on the features of the aforementioned instances. These highlight that generated images preserve the semantic content of the conditioning instance (a dog in the water, a bird with a long beak on a branch, and a landscape containing a water body) and present similarities with the real samples in the neighborhood of the instance. In cases such as the conditioning instance featuring a castle, the corresponding generated samples do not contain buildings; this could be explained by the fact that most of its neighbors do not contain castles either. Moreover, the generated images are not mere memorizations of training examples, as shown by the row of images immediately below, nor are they copies of the conditioning instance.

**Instance feature vector and noise interpolation.**    In Figure 6, we provide the resulting generated images when interpolating between the instance features of two data samples (vertical axis), shown on the left of each generated image grid, and additionally interpolating between two noise vectors in the horizontal axis. The top left quadrant shows generated images when interpolating between conditioning instance features from the class *husky*. The generated dog changes its fur color and camera proximity according to the instance conditioning. At the top right corner, when interpolating between two *mushroom* instance features, generated images change their color and patterns

Table 8: Results for ImageNet in terms of Precision (P) and Recall (R) [30] (bounded between 0 and 100), using 10,000 real and generated images. "Instance selection", only used for IC-GAN, indicates whether 1,000 conditioning instances are selected with k-means (k-means 1,000) or 10,000 conditioning instances are sampled uniformly (random 10,000) from the training set to obtain 10,000 generated images in both cases. *: Generated images obtained with the paper's opensourced code.

| Method | Res. | Instance selection | ↑P | ↑R |
|---|---|---|---|---|
| *Unlabeled setting* | | | | |
| Uncond. BigGAN | 64 | - | 69.6 ± 1.0 | 63.1 ± 0.0 |
| IC-GAN | 64 | k-means 1,000 | **74.2** ± 0.8 | 60.2 ± 0.6 |
| IC-GAN | 64 | random 10,000 | 67.5 ± 0.4 | **68.6** ± 0.5 |
| Self-cond. GAN [33]* | 128 | - | 66.3 ± 0.5 | 48.4 ± 0.8 |
| IC-GAN | 128 | k-means 1,000 | **78.2** ± 0.8 | 55.6 ± 0.9 |
| IC-GAN | 128 | random 10,000 | 71.7 ± 0.3 | **69.7** ± 0.9 |
| IC-GAN | 256 | k-means 1,000 | **77.7** ± 0.5 | 54.3 ± 0.7 |
| IC-GAN | 256 | random 10,000 | 70.4 ± 0.7 | **68.9** ± 0.3 |
| *Labeled setting* | | | | |
| BigGAN [5] | 64 | - | 72.8 ± 0.4 | 68.6 ± 0.6 |
| Class-conditional IC-GAN | 64 | k-means 1,000 | **76.6** ± 0.7 | 67.5 ± 0.8 |
| Class-conditional IC-GAN | 64 | random 10,000 | 69.6 ± 0.9 | **74.5** ± 0.8 |
| BigGAN [5] | 128 | - | **83.2** ± 0.7 | 64.2 ± 0.7 |
| Class-conditional IC-GAN | 128 | k-means 1,000 | 78.8 ± 0.3 | 64.3 ± 0.7 |
| Class-conditional IC-GAN | 128 | random 10,000 | 72.2 ± 0.4 | **73.6** ± 0.5 |
| BigGAN [5] | 256 | - | **83.9** ± 0.6 | 70.2 ± 0.7 |
| Class-conditional IC-GAN | 256 | k-means 1,000 | 82.2 ± 0.3 | 70.4 ± 0.3 |
| Class-conditional IC-GAN | 256 | random 10,000 | 73.9 ± 0.6 | **79.3** ± 0.2 |

Table 9: Results for ImageNet in unlabeled setting, comparing BigGAN and StyleGAN backbones. For fair comparison with [42] at $64 \times 64$ resolution, we trained an unconditional BigGAN model and report the non-official FID and IS scores – computed with Pytorch rather than TensorFlow – indicated with *. [†]: increased parameters to match IC-GAN capacity. DA: 50% horizontal flips in real and fake samples (**d**), and conditioning instances (**i**). $ch\times$: Channel multiplier that affects network width.

| Method | Res. | ↓FID | ↑IS |
|---|---|---|---|
| Uncond. BigGAN[†] | 64 | 16.9* ± 0.0 | 14.6* ± 0.1 |
| **StyleGAN2** + DA (**d**) | 64 | 12.4* ± 0.0 | 15.4* ± 0.0 |
| **IC-GAN (BigGAN)** + DA (**d,i**) | 64 | 9.2* ± 0.0 | **23.5*** ± 0.1 |
| **IC-GAN (StyleGAN2)** + DA (**d,i**) | 64 | **8.5*** ± 0.0 | **23.5*** ± 0.1 |
| Uncond. BigGAN [36] | 128 | 25.3 | 20.4 |
| **StyleGAN2** + DA (**d**) | 128 | 27.8 ± 0.1 | 18.8 ± 0.1 |
| **IC-GAN (BigGAN)** + DA (**d,i**) | 128 | **11.7** ± 0.0 | **48.7** ± 0.1 |
| **IC-GAN (StyleGAN2)** + DA (**d,i**) | 128 | 15.2 ± 0.1 | 38.3 ± 0.2 |
| **StyleGAN2** + DA (**d**) | 256 | 41.3 ± 0.1 | 19.7 ± 0.1 |
| **IC-GAN (BigGAN)** ($ch \times 64$) + DA (**d,i**) | 256 | 17.4 ± 0.1 | 53.5 ± 0.5 |
| **IC-GAN (BigGAN)** ($ch \times 96$) + DA (**d**) | 256 | **15.6** ± 0.1 | **59.0** ± 0.4 |
| **IC-GAN (StyleGAN2)** + DA (**d,i**) | 256 | 23.1 ± 0.1 | 42.2 ± 0.2 |

accordingly. Moreover, in the bottom left quadrant, *lorikeet* instance features are interpolated with flying *hummingbird* instance features, and the generated images change their color and appearance accordingly. Finally, in the bottom right grid, we interpolate instance features from a *tiger* and instance features from a *white wolf*, resulting in different blends between the striped pelt of the tiger and the white fur of the wolf.

**Unlabeled COCO-Stuff.** Training IC-GAN with a StyleGAN2 backbone on COCO-Stuff has resulted in quantitative results that surpass those achieved by the state-of-the-art LostGANv2 [49] and OC-GAN [50], controllable and conditional complex scene generation pipelines that rely on heavily labeled data (bounding boxes and class labels), tailored intermediate steps and somewhat complex architectures. In Figure 7, we compare generated images obtained with LostGANv2 and OC-GAN with those generated by IC-GAN with a StyleGAN2 backbone. Note that the two former methods use a bounding box layout with class labels as a conditioning, while we condition on the features extracted from the real samples $x_i$ depicted in Figure 7a. We compare the generations obtained with two random seeds for all methods, and observe that IC-GAN generates higher quality images in all cases, especially for the top three instances. Moreover, the diversity in the generations using two random seeds for LostGANv2 and OC-GAN is lower than for IC-GAN. This is not surprising, as the former methods are restricted by their bounding box layout conditioning that specifies the number of objects, their classes and their expected positions in the generated images. By contrast, IC-GAN conditions on an instance feature vector, which does not require any object label, cardinality or position to be satisfied, allowing more freedom in the generations.

**ImageNet.** Class-conditional IC-GAN with a BigGAN backbone has shown comparable quantitative results to those of BigGAN for $256 \times 256$ resolution in Subsection 3.4. In Figure 8, we can qualitatively compare BigGAN ($ch \times 64$) (first rows) and IC-GAN ($ch \times 64$) (second and third rows), for three class labels: *goldfish*, *limousine* and *red fox*. By visually inspecting the generated images, we can observe that the generation quality is similar for both BigGAN and IC-GAN in these specific cases. Moreover, IC-GAN allows controllability of the semantics by changing the conditioning instance features. For instance, changing the background in which the goldfish are swimming into lighter colors in Figure 8a, generating limousines in generally dark and uniform backgrounds or, instead, in an urban environment with a road and buildings (Figure 8b), or generating red foxes with a close up view or with a full body shot as seen in Figure 8c.

**Swapping classes for class-conditional IC-GAN on ImageNet.** In Figure 8, we show that we can change the appearance of the generated images by leveraging different instances of the same class. In Figure 9, we take a further step and condition on instance features from other classes. More specifically, in Figure 9 (top), we condition on the instance features of a snowplow in the woods surrounded by snow, and ask to generate snowplows, camels and zebras. Perhaps surprisingly, the generated images effectively get rid of the snowplow, and replace it by camel-looking and zebra-looking objects, respectively, while maintaining a snowy background in the woods. Moreover, when comparing the generated images with the closest samples in ImageNet, we see that for generated camels in the snow, the closest images are either a camel standing in dirt or other animals in the snow; for the generated zebras in the snow, we find one sample of a zebra standing in the snow, while others are standing in other locations/backgrounds. In Figure 9 (bottom), we condition on the features of an instance that depicts a golden retriever on a beach with overall purple tones, and ask to generate golden retrievers, camels or zebras. In most cases, generated images contain camels and zebras standing on water, while other generations contain purple or blue tones, similar to the instance used as conditioning. Note that, except one generated zebra image, the closest samples in ImageNet do not depict camels or zebras standing in the water nor on the beach.

# F  Additional off-the-shelf transfer results for IC-GAN

**Is IC-GAN able to shift the generated data distribution by conditioning on different instances?**
As discussed in Section 3.3, we can transfer an IC-GAN trained on unlabeled ImageNet to COCO-Stuff and obtain better metrics and qualitative results than with the same IC-GAN trained on COCO-Stuff. We hypothesize that the success of this experiment comes from the flexibility of our conditioning strategy, where the generative model exploits the generalization capabilities of the feature extractor when dealing with unseen instances to shift the distribution of generated images

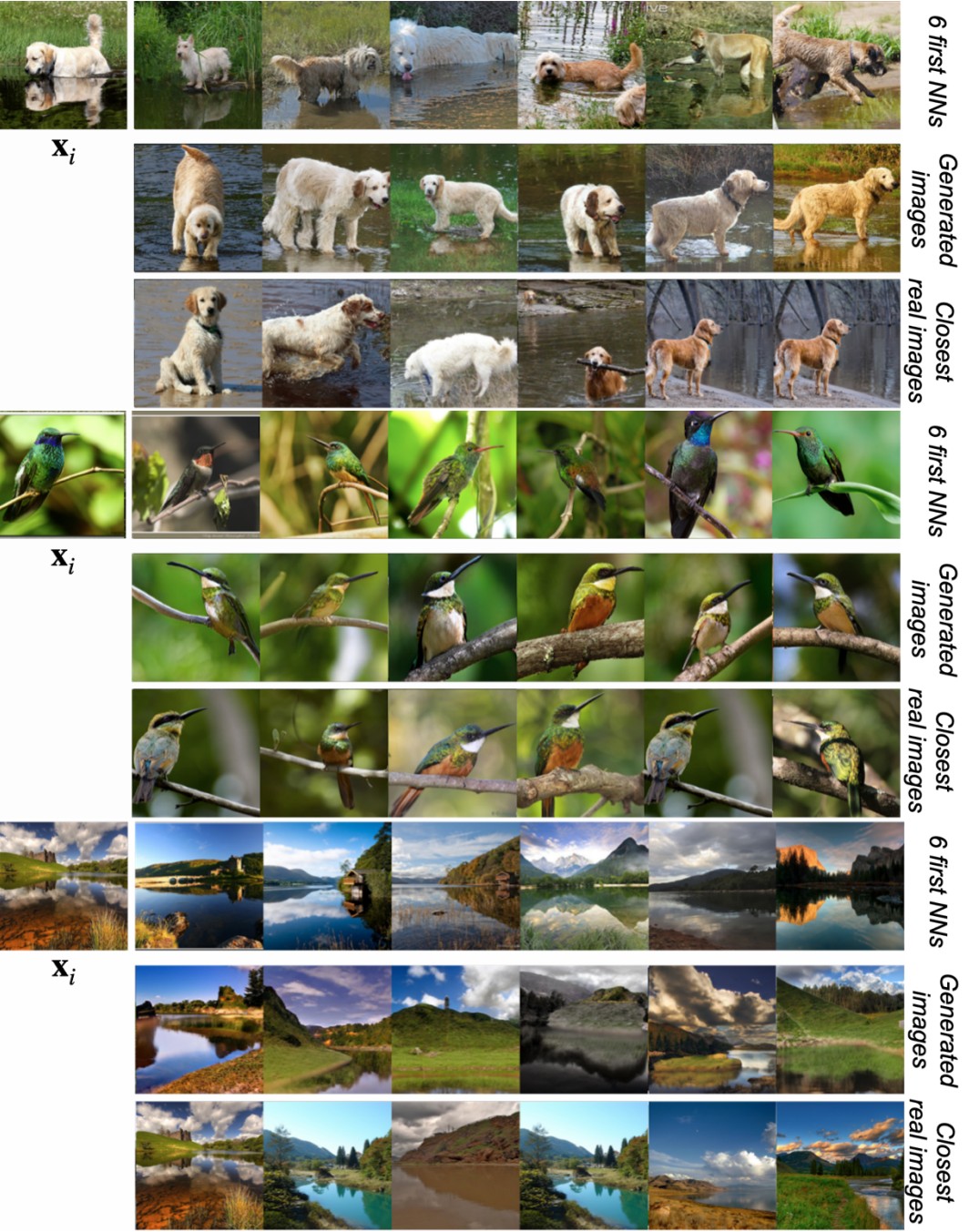

Figure 5: Qualitative results on unlabeled ImageNet ($256 \times 256$). Next to each input sample $\mathbf{x}_i$, used to obtain the instance features $\mathbf{h}_i = f_\theta(\mathbf{x}_i)$, the six nearest neighbors in the feature space of $f_\theta$ are displayed. Below the neighbors, generated images sampled from IC-GAN with a BigGAN backbone and conditioned on $\mathbf{h}_i$ are depicted. Immediately below the generated images, the closest image in the ImageNet training set is shown for each example (cosine distance in the feature space of $f_\theta$).

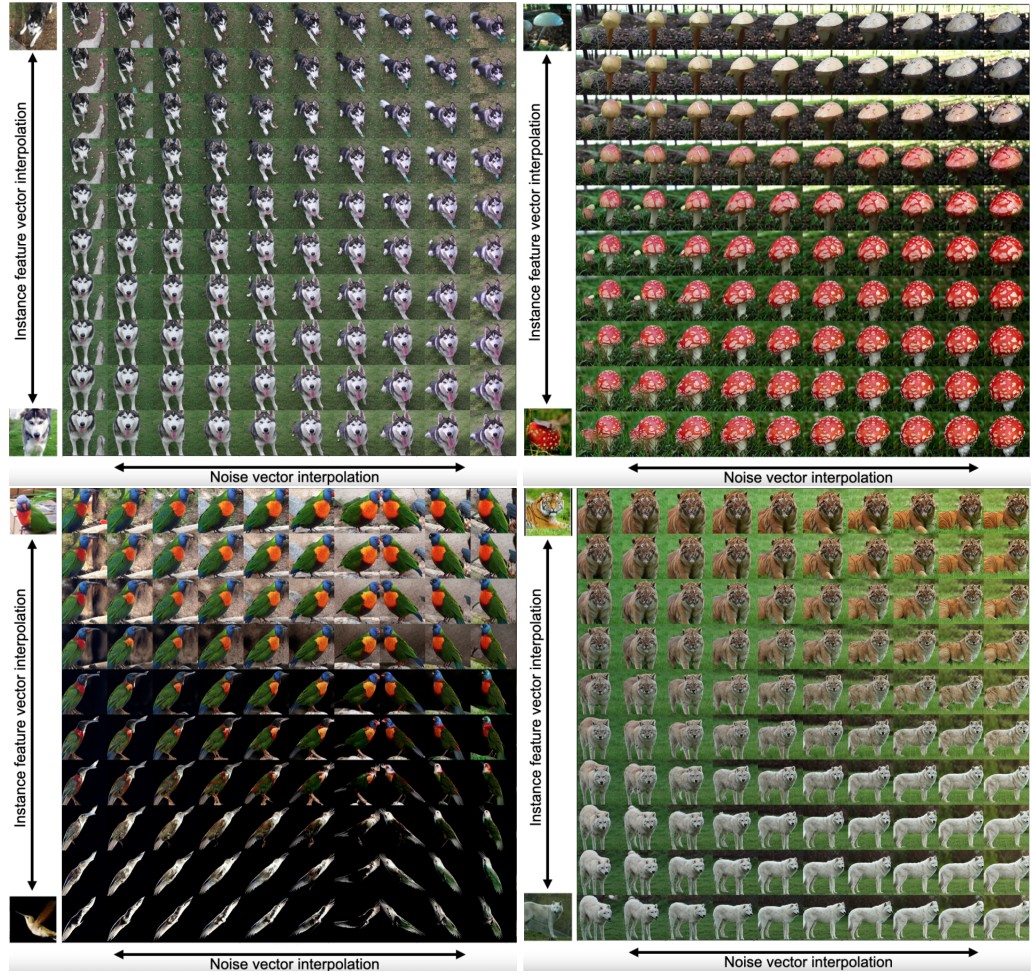

Figure 6: Qualitative results on unlabeled ImageNet ($256 \times 256$) using IC-GAN (BigGAN backbone) and interpolating between two instance feature vector conditionings (vertical axis) and two input noise vectors (horizontal axis). The two images depicted to the left of the generated image grids are used to extract the instance feature vectors used for the interpolation.

from ImageNet to COCO-Stuff. To test this hypothesis we design the following experiment: we compute FID scores of generated images obtained by conditioning IC-GAN with instance features from either ImageNet or COCO-Stuff and use either COCO-Stuff or ImageNet as a reference dataset to compute FID. In Table 10 (first row) we show that when using COCO-Stuff for both the instance features and the reference dataset, IC-GAN scores 8.5 FID; this is a lower FID score than the 43.6 FID obtained in Table 10 (second row) when conditioning IC-GAN on ImageNet instance features and using COCO-Stuff as reference dataset. Moreover, when using COCO-Stuff instance features and ImageNet as reference dataset, in Table 10 (third row), we obtain 37.2 FID. This shows that, by changing the conditioning instance features, IC-GAN successfully exploits the generalization capabilities of the feature extractor to shift the distribution of generated images to be closer to the COCO-Stuff distribution. Additionally, note that the distance between ImageNet and COCO-Stuff datasets can be quantified with an FID score of 37.2 [2].

**What is being transferred when IC-GAN is conditioned on instances other than the ones in the training dataset?** From the point of view of KDE, what is being transferred is the kernel shape, not the kernel location (that is controlled by instances). The kernel shape is predicted using a generative model from each input instance and we probe the kernel via sampling from the generator.

---

[2]We subsampled 76,000 ground-truth images from ImageNet training set and used all COCO-Stuff training ground-truth images.

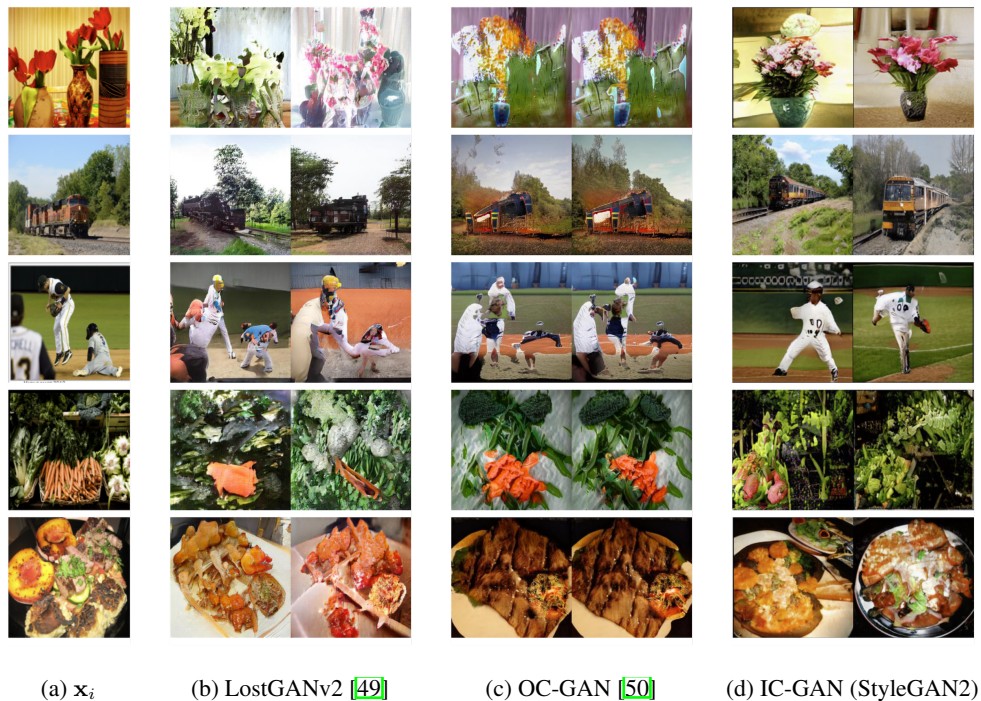

| (a) $\mathbf{x}_i$ | (b) LostGANv2 [49] | (c) OC-GAN [50] | (d) IC-GAN (StyleGAN2) |
|---|---|---|---|

Figure 7: Qualitative comparison for scene generation on $256 \times 256$ COCO-Stuff with other state-of-the-art scene generation methods. (a) Data samples $\mathbf{x}_i$ from which instance features $\mathbf{h}_i = f(\mathbf{x}_i)$ are obtained for IC-GAN, and labeled bounding box conditionings are obtained for LostGANv2 and OC-GAN. Images generated with two random seeds with (b) LostGANv2 [49], (c) OC-GAN [50], (d) IC-GAN (StyleGAN2).

Table 10: FID scores on COCO-Stuff $128 \times 128$, when using an IC-GAN trained on ImageNet and tested with instance features from either COCO-Stuff or ImageNet and using either of those datasets as reference. The metrics obtained by sampling 1,000 instance features (k-means) from either ImageNet or COCO, and generating 76,000 samples. As a reference, 76,000 real samples from COCO-Stuff or ImageNet training set are used.

|  | train instance dataset | eval instance dataset | FID reference dataset | ↓**FID** |
|---|---|---|---|---|
| IC-GAN | ImageNet | COCO-Stuff | COCO-Stuff | **8.5** ± 0.1 |
| IC-GAN | ImageNet | ImageNet | COCO-Stuff | 43.6 ± 0.1 |
| IC-GAN | ImageNet | COCO-Stuff | ImageNet | 37.2 ± 0.1 |

Thus, we transfer a function that predicts kernel shape from a conditioning, and this function seems to be robust to diverse instances as shown in the paper (e.g. see Figure 1c and 1d). Moreover, by visually inspecting the generated images in our transfer experiments, we observed that when transferring an IC-GAN trained on ImageNet to COCO-Stuff, if the model is conditioned on images that contain unseen classes in ImageNet, such as "giraffe", the model will still generate an animal that would look like a giraffe without the skin patterns and characteristic antennae, because ImageNet contains other animals to draw inspiration from. This suggests that the model generates plausible images that have some similar features to those present in the instance conditioning, but adapting it to the training dataset style. Along these lines, we also observed that in some cases, shapes and other object characteristics from one dataset are transferred to another (ImageNet to COCO-Stuff). Moreover, when we conditioned on instances from Cityscapes, the generated images were rather colorful, resembling more the color palette of ImageNet images rather than the Cityscapes one.

**Off-the-shelf transfer results for IC-GAN.** In Figure 10, we provide additional generated samples and their closest images in the ImageNet training set, when conditioning on unseen instance features from other datasets. Generated images often differ substantially from the closest image in ImageNet.

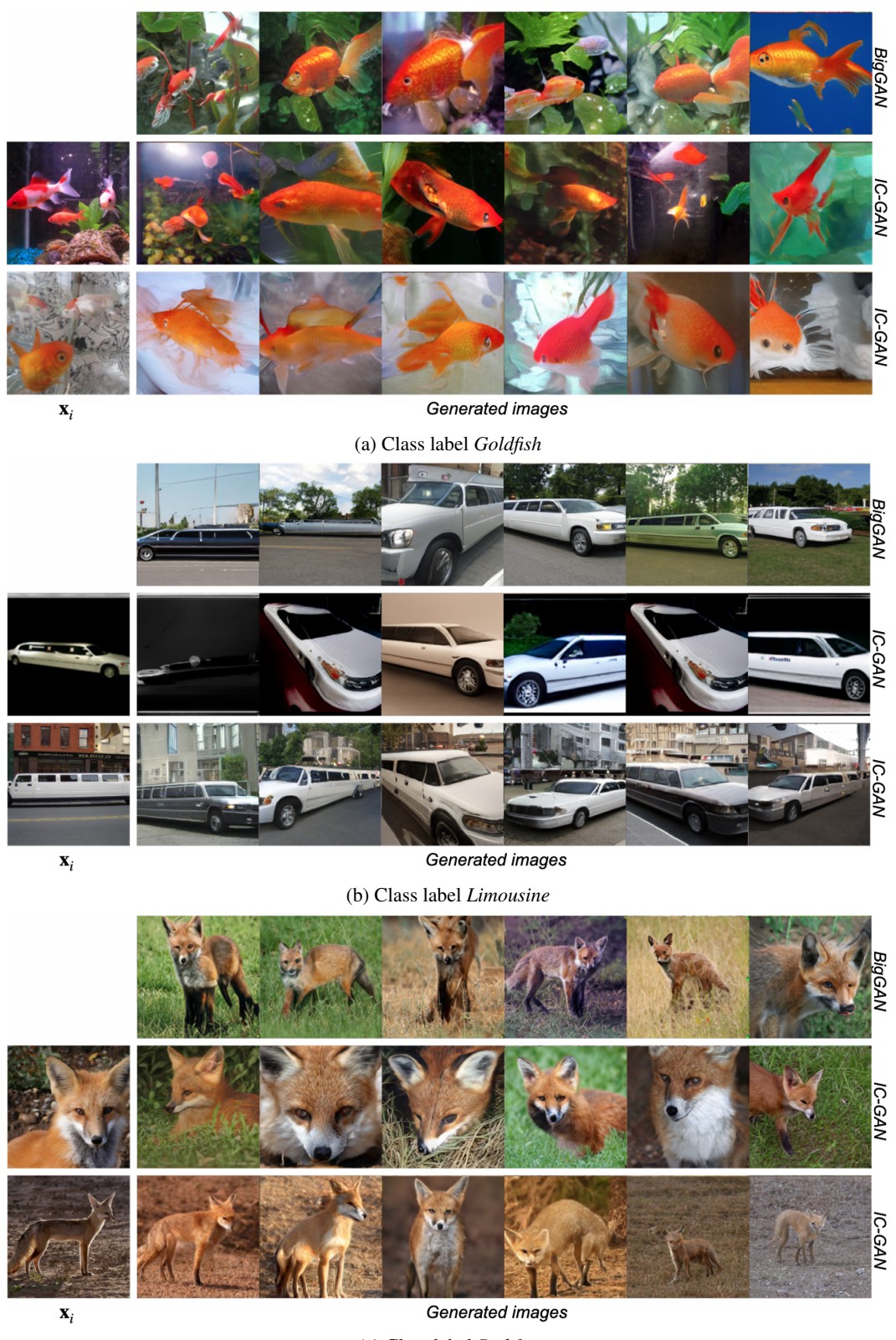

(a) Class label *Goldfish*

(b) Class label *Limousine*

(c) Class label *Red fox*

Figure 8: Qualitative results in $256 \times 256$ ImageNet. For each class, generated images with BigGAN are presented in the first row, while the second and third row show generated images using class-conditional IC-GAN with a BigGAN backbone, conditioned on the instance feature extracted from the data sample to their left ($\mathbf{x}_i$) and their corresponding class.

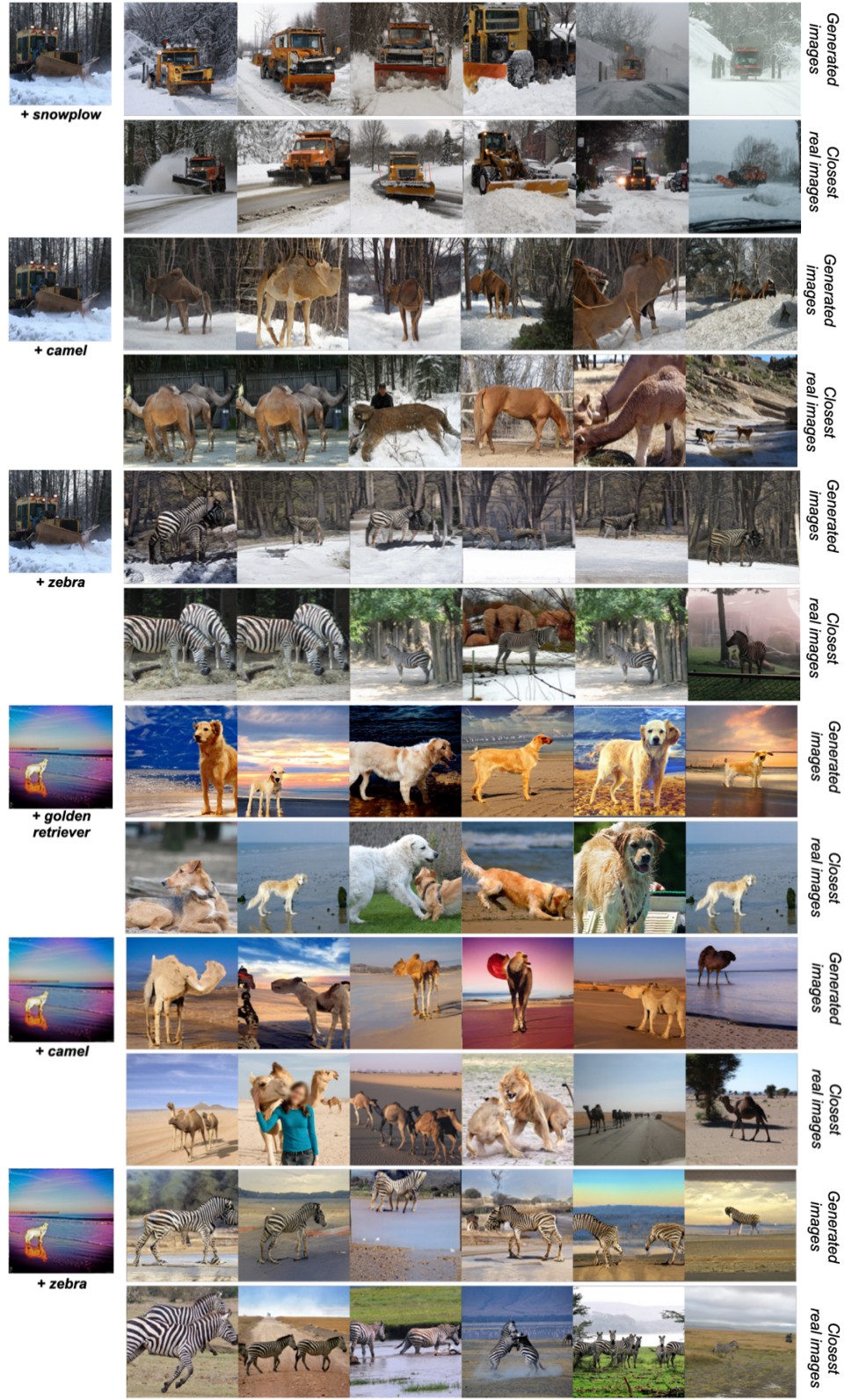

Figure 9: Generated $256 \times 256$ images with a class-conditional IC-GAN (BigGAN backbone) trained on ImageNet. Next to each data sample $\mathbf{x}_i$, used to obtain the instance features $\mathbf{h}_i = f_\theta(\mathbf{x}_i)$, we find generated images sampled from IC-GAN using $\mathbf{h}_i$ and six sampled noise vectors. Below the generated images, the closest image in the ImageNet training set are shown (Cosine similarity in the feature space of $f_\theta$).

Although generated images using a COCO-Stuff and Cityscapes instances may have somewhat similar looking images in ImageNet (for the first and second instances in Figure 10), the differences accentuate when conditioning on instance features from MetFaces, PACS or Sketch datasets, where, for instance, IC-GAN with a BigGAN backbone generates images resembling sketch strokes in the last row, even if the closest ImageNet samples depict objects that are not sketches.

**Off-the-shelf transfer results for class-conditional IC-GAN.** In Figure 11, we show additional results when transferring a class-conditional IC-GAN with a BigGAN backbone trained on ImageNet to other datasets, using an ImageNet class label but an unseen instance. We are able to generate camels in the grass by conditioning on an image of a cow in the grass from COCO-Stuff, we show generated images with a zebra in an urban environment by conditioning on a Cityscapes instance, and we generate cartoon-ish birdhouses by conditioning on a PACS cartoon house instance. This highlights the ability of class-conditional IC-GAN to transfer to other datasets *styles* while maintaining the class label semantics.

## G   Class balancing in ImageNet-LT

We experimented with class balancing for BigGAN in the ImageNet-LT dataset. In Table 11, we compare (1) **BigGAN**, where both the class distribution for the generator and the data for the discriminator are long-tailed; (2) **BigGAN (CB)**, a class-balanced version of BigGAN, where the generator samples class labels from a uniform distribution and the samples fed to the discriminator are also class-balanced; and (3) **BigGAN (T = 2)** where the class distribution is balanced with a softmax temperature of T = 2 providing a middle ground between the long-tail and the uniform distributions. In the latter case, the probability to sample class $c$ (with a frequency $f_c$ in the long-tailed training set) during training is given by $p_c = \text{softmax}(T^{-1} \ln f_c)$.

Interestingly, balancing the class distribution (either with uniform class distribution or with T=2) harmed performance in all cases except for the validation Inception Score. We hypothesize that over-sampling rare classes, and thus the few images therein, may result in overfitting for the discriminator, leading to low quality image generations.

Table 11: ImageNet-LT quantitative results for different class balancing techniques. "t.": training and "v.": validation.

| | Res. | ↓t. FID | ↑t. IS | ↓v. FID | ↓v. [many/med/few] FID | ↑v. IS |
|---|---|---|---|---|---|---|
| BigGAN | 64 | **27.6** ± 0.1 | **18.1** ± 0.2 | **28.1** ± 0.1 | **28.8/32.8/48.4** ± 0.2 | 16.0 ± 0.1 |
| BigGAN (CB) | 64 | 62.1 ± 0.1 | 10.7 ± 0.2 | 56.2 ± 0.1 | 62.2/59.7/74.7 ± 0.2 | 11.0 ± 0.0 |
| BigGAN (T = 2) | 64 | 30.6 ± 0.1 | 16.8 ± 0.1 | 29.2 ± 0.1 | 30.9/33.3/49.5 ± 0.2 | **16.4** ± 0.1 |

## H   Choice of feature extractor

We study the choice of the feature extractor used to obtain instance features in Table 12. We compare results using an IC-GAN with a BigGAN backbone when coupling it with a ResNet50 feature extractor trained with either self supervision (SwAV) or with supervision for classification purposes (RN50) on ImageNet dataset. Results highlight similar IC-GAN performances for both feature extractors, suggesting that the choice of feature extractor that does not greatly impact the performance of our method when leveraging unlabeled datasets. Given that for the unlabeled scenario we assume no labels are available, we use the SwAV feature extractor. However, in the class-conditional case, we observe that the IC-GAN coupled with a RN50 feature extractor surpasses IC-GAN coupled with a SwAV feature extractor. Therefore, we choose the RN50 feature extractor for the class-conditional experiments. For ImageNet-LT, we transfer these findings and use a RN50 trained on ImageNet-LT as feature extractor, assuming we do not have access to the entire ImageNet dataset and its labels.

## I   Number of conditioning instance features at train time

To demonstrate that using many fine-grained overlapping partitions results in better performance than using a few coarser partitions, we trained IC-GAN with a BigGAN backbone by conditioning on all 1.2M training instance features at training time in ImageNet and a neighborhood size of $k = 50$, and compared it quantitatively with an IC-GAN trained by conditioning on only 1,000 instance features

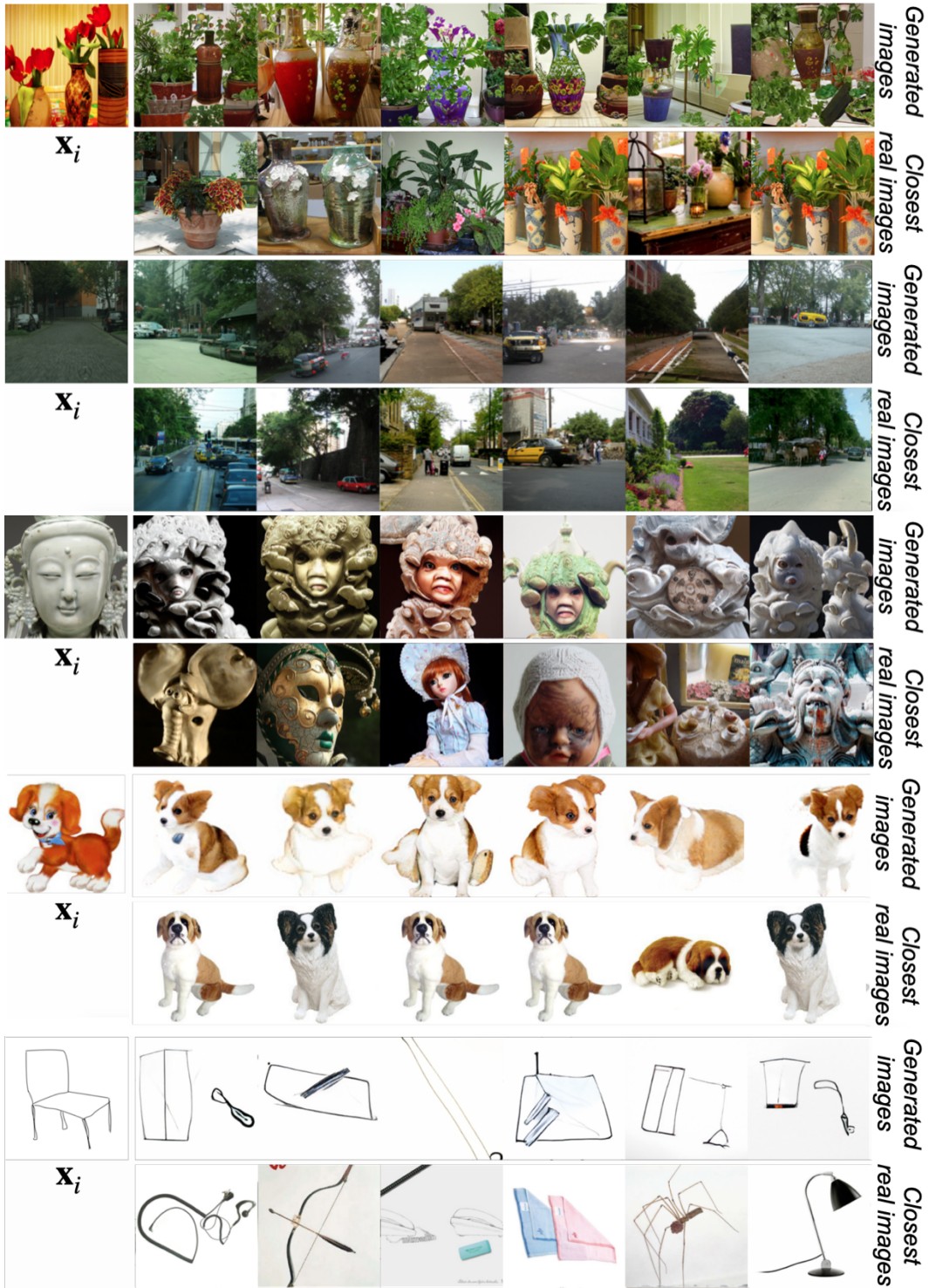

Figure 10: Qualitative off-the-shelf transfer results in $256 \times 256$, using an IC-GAN trained on unlabeled ImageNet and conditioning on unseen instances from other datasets. The instances come from the following datasets (top to bottom): COCO-Stuff, Cityscapes, MetFaces, PACS (cartoons), Sketches. Next to each data sample $\mathbf{x}_i$, used to obtain the instance features $\mathbf{h}_i = f_\theta(\mathbf{x}_i)$, generated images conditioning on $\mathbf{h}_i$ are displayed. Immediately below each generated image, the closest image in the ImageNet training set is displayed (Euclidean distance in the feature space of $f_\theta$).

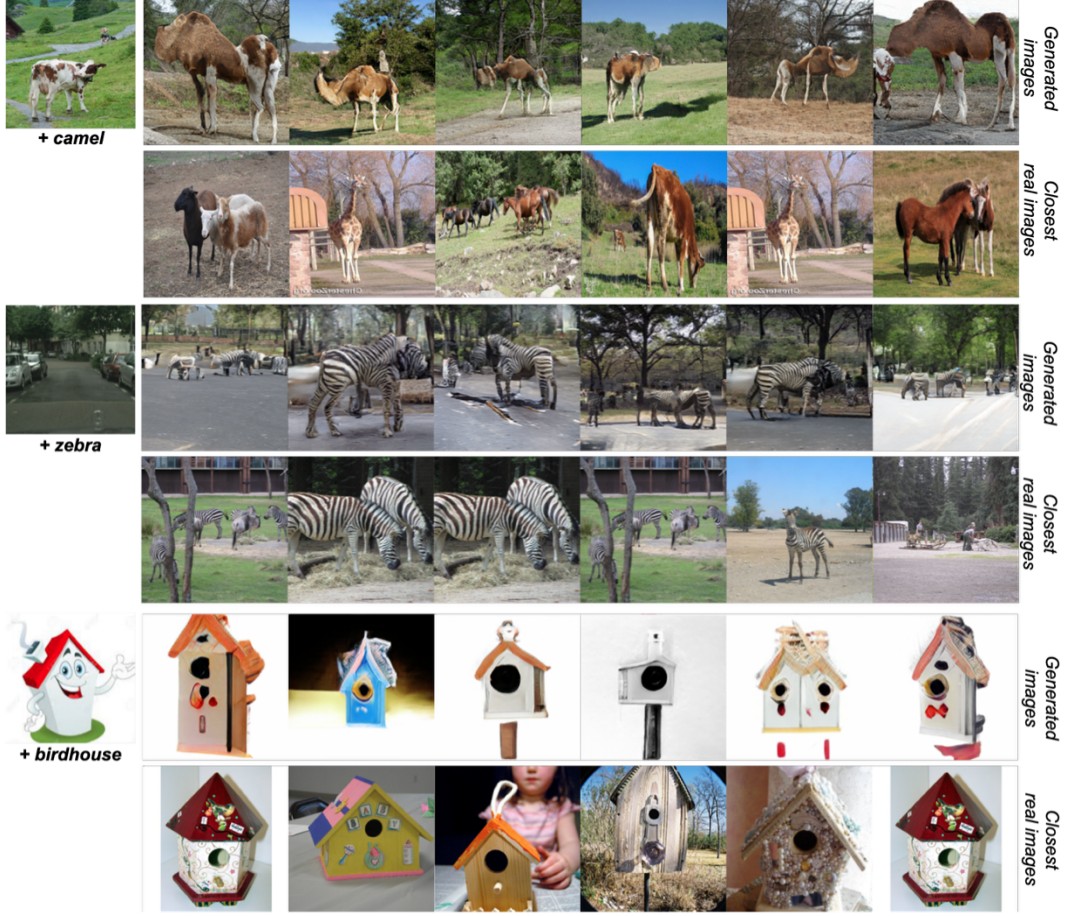

Figure 11: Qualitative off-the-shelf transfer results in $256 \times 256$, using a class-conditional IC-GAN trained on ImageNet and conditioning on unseen instances from other datasets and a class label. The instances come from the following datasets (top to bottom): COCO-Stuff, Cityscapes, PACS (cartoons). On the left, a data sample $\mathbf{x}_i$ is depicted, used to obtain the instance features $\mathbf{h}_i = f_\theta(\mathbf{x}_i)$. Next to the data samples, generated images conditioning on $\mathbf{h}_i$ and a class label (under the data samples) are displayed. Just below the generated images, the closest image in the ImageNet training set are shown (Euclidean distance in the feature space of $f_\theta$).

at training time. In this case, we extend the neighborhood size to $k = 1{,}200$ to better cover the training distribution [3]. Note that using $k = 50$ would result in using at most 50,000 training samples during training, an unfair comparison. The 1,000 instance features are selected by applying k-means to the entire ImageNet training set. We then use the same instances to generate images. Results are presented in Table 13 and emphasize the importance of training with all available instances, which results in significantly better FID and IS presumably due to the increased number of conditionings and their smaller neighborhoods.

## J  Matching storage requirements for IC-GAN and unconditional models

We hypothesize that the good performance of IC-GAN on ImageNet and COCO-Stuff can not solely be attributed to the slight increase of parameters and the extra memory required to store the instance features used at test time, but also to the IC-GAN design, including the finegrained overlapping partitions and the instance conditionings. To test for this hypothesis, we performed experiments with the unconditional BigGAN baseline on ImageNet and COCO-Stuff, training it by setting all

---

[3]Note that this setup resembles the class partition in ImageNet, where 1,000 classes contain approximately 1,200 images each.

Table 12: Feature extractor impact with SwAV (ResNet50 trained with a self-supervised approach) and RN50 (ResNet50 trained for the classification task in ImageNet). Experiments performed in $64 \times 64$ ImageNet, using $1,000$ training instance features at test time, selected with k-means.

|  | ↓**FID** |
| --- | --- |
| IC-GAN + SwAV | $11.7 \pm 0.1$ |
| IC-GAN + RN50 | $\mathbf{11.3} \pm 0.1$ |
| Class-conditional IC-GAN + SwAV | $9.9 \pm 0.1$ |
| Class-conditional IC-GAN + RN50 | $\mathbf{8.5} \pm 0.0$ |

Table 13: Comparison between training IC-GAN (BigGAN backbone) using only $1,000$ conditioning instance features (selected with k-means) or all training instance features during training, in IC-GAN $64 \times 64$. At test time, we condition IC-GAN on $1,000$ training instance features, selected with k-means.

| Method | ↓**FID** | ↑**IS** |
| --- | --- | --- |
| IC-GAN ($k = 50$, trained with all 1.2M conditionings) | $\mathbf{11.7} \pm 0.1$ | $\mathbf{21.6} \pm 0.1$ |
| IC-GAN ($k = 1,200$, trained with only 1,000 conditionings) | $24.8 \pm 0.1$ | $16.4 \pm 0.1$ |

labels in the training set to zero, following [36, 42], and increasing the generator capacity such that it matches the IC-GAN storage requirements. In particular, we not only endowed the unconditional BigGAN with additional parameters to compensate for the capacity mismatch, but also for the instances required by IC-GAN. Moreover, we performed analogous experiments for the unconditional StyleGAN2 in COCO-Stuff.

**ImageNet.** Given its instance conditioning, the IC-GAN (BigGAN backbone) generator introduces an additional 4.5M parameters when compared to the unconditional BigGAN generator. Moreover, IC-GAN requires an extra 8MB to store the $1,000$ instance features used at inference time. This 8MB can be translated into roughly 2M parameters[4]. Therefore, we compensate for this additional storage in IC-GAN by increasing the unconditional BigGAN capacity by expanding the width of both generator and discriminator hidden layers. We follow the practice in [5], where the generator and discriminator's width are changed together. The resulting unconditional BigGAN baseline generator has an additional 6.5M parameters. Results are reported in Table 14, showing that adding extra capacity to the unconditional BigGAN leads to an improvement in terms of FID and IS. However, IC-GAN still exhibits significantly better FID and IS, highlighting that the improvements cannot be solely attributed to the increase in parameters nor instance feature storage requirements.

**COCO-Stuff.** Similarly, IC-GAN (BigGAN backbone) trained on COCO-Stuff requires 4M additional parameters on top of the extra storage required by the $1,000$ stored instance features (8MB again translated into roughly 2M parameters). Therefore, we add 6M extra parameters to the unconditional BigGAN generator. In the case of IC-GAN with a StyleGAN2 backbone, the instance feature conditionings constitute 1M additional parameters. We therefore increase the capacity of the unconditional StyleGAN2 model by 3M to match the storage requirements of IC-GAN (StyleGAN2 backbone). The results are presented in Table 15, where it is shown that both the unconditional BigGAN and unconditional StyleGAN2 do not take advantage of the additional capacity and achieve poorer performance than the model with lower capacity, possibly due to overfitting. When compared to IC-GAN, the results match the findings in the ImageNet dataset: IC-GAN exhibits lower FID when using BigGAN or StyleGAN2 backbones, compared to their respective unconditional models with the same storage requirements, further highlighting that IC-GAN effectively benefits from its design, finegrained overlapping partitions, and instance conditionings.

## K  Additional neighborhood size impact studies

We additionally study the impact of the neighborhood size for ImageNet-LT in Table 16 and in COCO-Stuff in Table 17, showing that in both cases, IC-GAN with a BigGAN backbone and $k = 5$

---

[4]We store both parameters and instance features as float32.

Table 14: Comparing IC-GAN with the unconditional counterparts of BigGAN on $64 \times 64$ ImageNet with the same storage requirements. Storage-G counts the storage required for the generator, Storage-I the storage required for the training instance features, and Storage-All is the sum of both generator and instance features required storage. FID and IS scores are computed using Pytorch code.

| Method | #prms. | Storage-G | Storage-I | Storage-All | ↓FID | ↑IS |
|---|---|---|---|---|---|---|
| Unconditional BigGAN | 32.5M | 124MB | 0MB | 124MB | $30.0 \pm 0.1$ | $12.1 \pm 0.1$ |
| Unconditional BigGAN | 39M | 149MB | 0MB | 149MB | $16.9 \pm 0.0$ | $14.6 \pm 0.1$ |
| IC-GAN (BigGAN) | 37M | 141MB | 8MB | 149MB | $\mathbf{10.4} \pm 0.1$ | $\mathbf{21.9} \pm 0.1$ |

Table 15: Comparing IC-GAN with the unconditional counterparts on $128 \times 128$ COCO-Stuff, with the same storage requirements. Storage-G counts the storage required for the generator, Storage-I the storage required for the training instance features, and Storage-All is the sum of both generator and instance features required storage.

| | #prms. | Storage-G | Storage-I | Storage-All | ↓FID | |
|---|---|---|---|---|---|---|
| | | | | | train | eval |
| Unconditional BigGAN | 18M | 68MB | 0MB | 68MB | $17.9 \pm 0.1$ | $46.9 \pm 0.5$ |
| Unconditional BigGAN | 24M | 92MB | 0MB | 92MB | $28.8 \pm 0.1$ | $58.1 \pm 0.5$ |
| IC-GAN (BigGAN) | 22M | 84MB | 8MB | 92MB | $\mathbf{16.8} \pm 0.1$ | $\mathbf{44.9} \pm 0.5$ |
| Unconditional StyleGAN2 | 23M | 88MB | 0MB | 88MB | $8.8 \pm 0.1$ | $37.8 \pm 0.2$ |
| Unconditional StyleGAN2 | 26M | 100MB | 0MB | 100MB | $9.4 \pm 0.0$ | $38.4 \pm 0.3$ |
| IC-GAN (StyleGAN2) | 24M | 92MB | 8MB | 100MB | $\mathbf{8.7} \pm 0.0$ | $\mathbf{35.8} \pm 0.1$ |

achieves the best FID and IS metrics. The choice of a lower neighborhood size $k = 5$ than in the ImageNet case ($k = 50$) could suggest that the number of semantically similar neighboring samples is smaller for these two datasets. This wouldn't be completely surprising given that these two datasets are considerably smaller than ImageNet. Increasing the value of $k$ in COCO-Stuff and ImageNet-LT would potentially gather samples with different semantics within a neighborhood, which could potentially harm the training and therefore the generated images quality.

Finally, in Figure 12, we qualitatively show generated images in ImageNet when using an IC-GAN trained with varying neighborhood sizes. The findings further support the ones presented in Subsection 3.5, showing that smaller neighborhoods result in generated images with less diversity, while bigger neighborhood sizes, for example $k = 500$ result in more varied but lower quality generations, supporting that $k$ controls the smoothing effect.

Table 16: Impact of the number of neighbors ($k$) used to train class-conditional IC-GAN (BigGAN backbone) in ImageNet-LT $64 \times 64$. Reported results in ImageNet validation set. As a feature extractor, a ResNet50 is trained as a classifier on the same dataset is used. 50k instance features are sampled from the training set.

| | ↓FID | ↑IS |
|---|---|---|
| $k = 5$ | $\mathbf{23.4} \pm 0.1$ | $\mathbf{17.6} \pm 0.1$ |
| $k = 20$ | $24.1 \pm 0.1$ | $16.8 \pm 0.1$ |
| $k = 50$ | $24.1 \pm 0.1$ | $16.7 \pm 0.1$ |
| $k = 100$ | $25.6 \pm 0.1$ | $16.3 \pm 0.1$ |
| $k = 500$ | $27.1 \pm 0.1$ | $15.3 \pm 0.1$ |

Table 17: Impact of the number of neighbors ($k$) used to train IC-GAN (BigGAN backbone) on COCO-Stuff $128 \times 128$. Reported results on COCO-Stuff evaluation set. As a feature extractor, a ResNet50 trained with self-supervision (SwAV) is used.

|        | ↓**FID**        |
| ------ | --------------- |
| $k=5$   | **44.9** $\pm$ 0.5 |
| $k=20$  | 46.8 $\pm$ 0.3  |
| $k=50$  | 45.8 $\pm$ 0.4  |
| $k=100$ | 48.4 $\pm$ 0.3  |
| $k=500$ | 48.3 $\pm$ 0.5  |

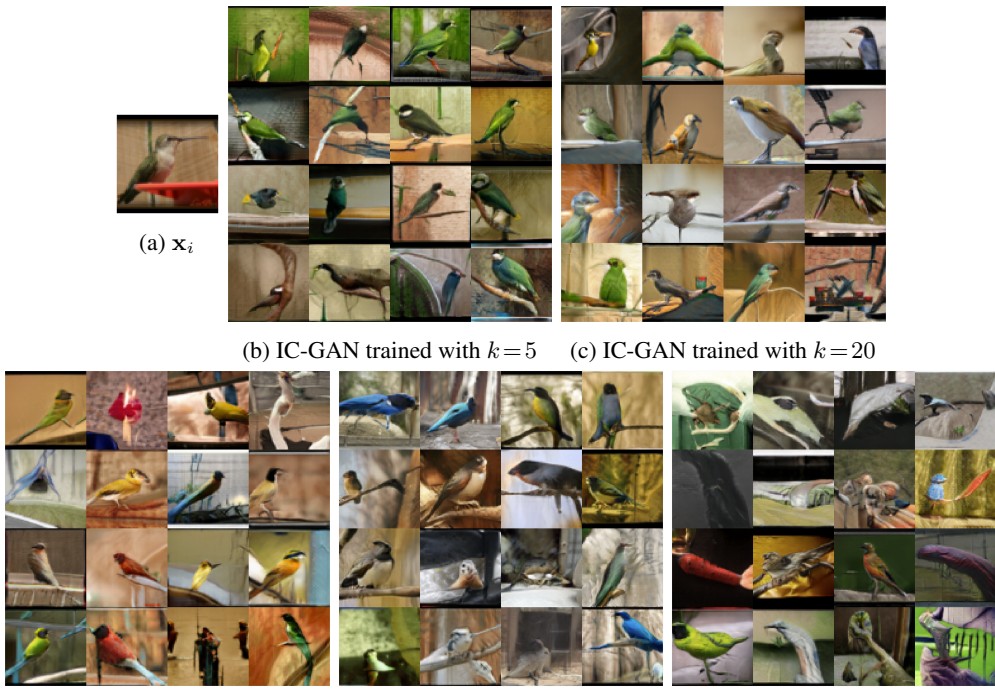

(a) $\mathbf{x}_i$

(b) IC-GAN trained with $k=5$    (c) IC-GAN trained with $k=20$

(d) IC-GAN trained with $k=50$   (e) IC-GAN trained with $k=100$   (f) IC-GAN trained with $k=500$

Figure 12: Qualitative results in $64 \times 64$ unlabeled ImageNet when training IC-GAN (BigGAN backbone) with different neighborhood sizes $k$. (a) Data samples $\mathbf{x}_i$ used to obtain the instance features $\mathbf{h}_i = f_\theta(\mathbf{x}_i)$. (b-f) Generated images with IC-GAN (BigGAN backbone), sampling different noise vectors, for different neighborhood sizes $k$ used during training.