# OpenReview forum: "Instance-Conditioned GAN"
_NeurIPS.cc/2021/Conference — NeurIPS 2021 Spotlight_

### Official Review · Reviewer_MiBY · 2021-07-06

**Rating:** 5
**Confidence:** 4

**Summary:**

To model complex distributions of datasets, this paper proposed to partition datasets into a mixture of overlapping neighborhoods described by a datapoint and its nearest neighbors. An instance-conditioned generator is trained to learn the distribution around each datapoint, namely the generator learns to generate images similar to the conditional images. The proposed approach is evaluated on ImageNet and Coco-Stuff.

**Limitations And Societal Impact:**

Please refer to Cons in the main review sections.

**Main Review:**

- Pros:
    - The paper is well written and easy to read.
    - Comprehensive experiments are conducted, and multiple metrics are adopted to evaluate the proposed method.

- Cons:
    - The contribution is marginal. The authors claim the proposed method can learn complex distributions, however the generator can only generate images similar to the given conditional image, and the the number of modes to display during test phase largely depends on the conditional images provided.
    - Can author provide several real-life applications when this model could be helpful?
    - Is the proposed approach indeed under unconditional setting? It is more like image-to-image translations, instead of unconditional generations.

**Time Spent Reviewing:**

2

---

> ### Author Response · Authors · 2021-08-09
> **Answer to reviewer MiBY**
>
> We would like to thank the reviewer for their useful feedback and comments. We are encouraged that the reviewer found the paper well written and easy to read and the experimental section to be comprehensive. We address the reviewer’s comments below and will include all the feedback in the revised version of the manuscript.
>
>
> - **“The contribution is marginal.”**  We respectfully disagree. As noted by other reviewers, the approach is novel [Reviewer **pxsk**], and the idea is original and significant [Reviewer **8u3S**]. The paper proposes a novel approach that takes inspiration from KDE and models dataset distributions in a non-parametric way by conditioning both generator and discriminator on instance features. This formulation sets a new state of the art in image generation when no labels are available, exhibiting high quality results when generating single object images (ImageNet) as well as complex scenes (COCO-stuff). In the class-conditional setting, the model allows for controllable image generation and achieves competitive results which surpass the baselines when learning from long-tailed data distributions. Finally, for the first time, we show how a generative model can effortlessly transfer to datasets not seen during training.
>
> - **“The authors claim the proposed method can learn complex distributions, however the generator can only generate images similar to the given conditional image...”**  Not quite. We note that the data distribution learned by the generator may not be as simple as the reviewer suggests. Given the GAN training objective, the distribution in image space learned around each datapoint may be complex and have more than one mode. Moreover, we would like to emphasize that the generated images will be similar in the feature space, but may result in notable differences in the image space. We invite the reviewer to inspect the visual results in Figure 5 and 8 in the Supplementary Material. Nevertheless, it is the intended behavior of the model to generate samples that are somewhat similar to the conditioning instance. Despite being similar, we have shown that they are diverse and different from the real neighbors (see Figure 5 in the Supplementary material).
>
> - **“... and the number of modes to display during test phase largely depends on the conditional images provided.”** It is true that the number of modes at evaluation time might be correlated with the number of conditionings provided; however, let us provide a bit of context to the reviewer’s statement. Given the GAN objective, different conditionings may lead to different numbers of modes. Moreover, we would like to note that this correlation is characteristic of non-parametric density estimators such as KDE, where the number of modes depends on both the number of instances and the kernel bandwidth. Intuitively, in our case, we control the kernel bandwidth with the number of neighbors of an instance during training.
>
> - **“Can author provide several real-life applications when this model could be helpful?”**  IC-GAN can be used in all cases where (class-conditional) GANs can be used, and provides improved image diversity and controllability for those cases, while maintaining or improving on the visual quality. In addition, IC-GAN, as a data generative model, is particularly useful for the cases mentioned below:
>     * By generating similar but diverse samples given a conditioning instance, we can move away from standard data augmentation techniques, which are crucial for both supervised classification tasks and self-supervised methods, and use the generated samples from IC-GAN as augmentation instead.
>     * It can help mitigate dataset biases by generating images of under-represented instances.
>     * The class-conditional results can be beneficial to help mitigate spurious correlations in a classification task. For example, by generating camels on the snow, zebras in urban scenes, or camels in the ocean, we can easily break dataset correlations which lead to unexpected behaviors in image classification tasks.
>     * For artistic applications, IC-GAN provides more controlled generations due to the use of instances. A user or an artist may want to control the kind of generations that a model gives them, and IC-GAN enables this control through its instance conditioning (and potentially a class label).
>
>
> - **“Is the proposed approach indeed under unconditional setting?“**  No, IC-GAN is NOT an unconditional method (it is conditioned on feature vectors). However, it does not require any labels at training time. Note that this is different in the class-conditional case. As pointed out in the introduction, none of the state of the art clustering-based approaches to unlabeled image generation is truly unconditional, as a cluster index is required to condition the model.
> - **"It is more like image-to-image translations, instead of unconditional generations".** Image to image translation has a target domain. However, in the case of IC-GAN, we do not attempt to translate images from one domain to another, but rather generate images that look like real image neighbors of each conditioning instance.

---

### Official Review · Reviewer_8u3S · 2021-07-16

**Rating:** 7
**Confidence:** 4

**Summary:**

The authors propose a new way of training GANs, which they call instance-conditioned GANs. This method is similar to conditional GANs, but instead of using a label, they use a feature vector extracted from some feature function (by using ResNet50 in the experiments).This bypasses the need for labels for conditional GANs. However, there is another difference. The features are assumed to be locally similar so when evaluating the loss over batches, real images are chosen to be neighbors of the given instance. This creates a kernel-density estimation type of model. Experimentally, these Instance-condition GANs perform very well, beating self-supervised, and unconditional GANs. Moreover, the conditional version of instance-conditioned GANs outperform conditional GANs as well in some smaller-resolution settings.

**Limitations And Societal Impact:**

Limitations:

The embedding depends on a pretrained network.

Potential negative societal impacts:

None that I can think of

**Main Review:**

Originality:

This paper is very original and interesting.

Quality:

The paper is a bit confusing (see clarity below) but the research does seem to be well developed. The experiments are very impressive as well!

Clarity:

The overall clarity of this paper is only okay. Some of the terms are a bit confusing. For example, on line 71, the authors say “overlapping partitions”. What are overlapping partitions of a manifold? Typically, given a set A, we partition it. Meaning that there is a singular partition of disjoint sets whose union is the whole set A. I can only conclude that overlapping partitions means a collection of sets whose union is A but they are not disjoint.

It is also confusing why they call a datapoint an instance. In fact this is defined on line 72 but used without definition on line 38, causing more confusion.

Line 79 says that Figure 2a shows 7 neighbors which is confusing because there are only 4 images, although there are 7 darker shapes. And the different shapes in the figure make it also confusing because one would think that neighboring shapes be similar.

Significance:

This is a very interesting idea and significant because it can be used on completely unsupervised data.

**Time Spent Reviewing:**

3

---

> ### Author Response · Authors · 2021-08-09
> **Answer to reviewer 8u3S**
>
> We would like to thank the reviewer for their thoughtful feedback and suggestions to improve the clarity of the paper. We are encouraged that the reviewer found the paper very original and interesting, the idea interesting and significant, the research well developed, and the experiments impressive. We thank the reviewer for their strong support of our work. We address the reviewer’s comments below and will include all the feedback in the revised version of the manuscript.
>
>
> - **Clarity of the presentation**. While it is not possible to upload an updated manuscript just yet, we provide a list of changes -- following the reviewer’s suggestions --  that will hopefully improve the clarity of the paper.
>     - Line 71: We have changed “*overlapping partitions*” to “*overlapping clusters*”, where the word cluster refers to a group of points close to each other and close to the instance conditioning features. We hope that this will make the terminology more clear.
>     - Line 38: We have rephrased the sentence to make it clear from the start that “*instance*” here refers to a “*data point*”:  “*More precisely, IC-GAN learns to model the distribution of the neighborhood of a data point, also referred to as instance, by  providing  a  representation  of  the  instance  as  an  additional  input…*”.
>     - Figure 2: We have changed the caption of the figure to clarify that only five out of eight neighboring images are shown (to avoid cluttering the figure too much), and that different shapes correspond to different class labels, as samples in a neighborhood could belong to different classes: “*The goal of the generator is to generate realistic images similar to the neighbors of*  $\mathbf{h}_i$,  *defined in the embedding space using cosine similarity. Five out of eight neighbors are shown in the figure.* *Note that images in the same neighborhood may belong to different classes* *(depicted as different shapes)*.”
> - **Limitations**. Following the reviewer’s suggestion, we have added the following sentence in the Discussion section (Limitations paragraph) : “*Second, the instance feature vectors used to condition the model are obtained with a pre-trained feature extractor (self-supervised in the unlabeled case) and depend on it. We speculate that this limitation might be mitigated if the feature extractor and the generator are trained jointly, and leave it as future work*.””

---

> > ### Comment · Reviewer_8u3S · 2021-08-24
> > **Thank you for the response**
> >
> > Thank you for the response and effort in making the paper clearer and more specific. I will maintain my original score as it is a good paper.

---

### Official Review · Reviewer_pxsk · 2021-07-16

**Rating:** 7
**Confidence:** 4

**Summary:**

This paper introduces Instance Conditioned GAN (IC-GAN) which aims to model complex multi modal distributions in an unconditional manner. The model partitions the target distribution into sub distributions learned by conditioning on a single training point and its nearest neighbors. IC-GAN improves unconditional image performance baselines on ImageNet and COCO-stuff with a variety of architectures. The authors extend the model to perform class conditional generation and transfer learning.

**Limitations And Societal Impact:**

I think the authors have adequately addressed limitations and societal impact of their work.

**Main Review:**

Unconditional generation for datasets like ImageNet is a very hard problem and this paper proposes a novel way of partitioning the dataset. The authors draw parallels to Kernel density estimation (KDE) to motivate the approach which to my knowledge has not been done. The ease of performing transfer learning adds to the novelty of the approach.

The paper is very well written and easy to understand. The authors have conducted a thorough set of experiments and ablation studies to back up their claims.

Some observations:
1. The authors claim their method produces a diverse set of images but they do not share recall values anywhere apart from Fig 4. I think adding some metric of diversity in addition to FID/IS to at least some experiments would help.
2. In the transfer learning section, I would have liked a little bit more discussion about what the authors think is being transferred. For example, if we condition on an image from a different dataset, what property of that dataset its discarded and what what property is preserved/ transferred?

In all, I think the authors have performed a comprehensive set of experiments which help push forward our understanding in unconditional generation and, therefore, I would recommend acceptance.

**Time Spent Reviewing:**

3

---

> ### Author Response · Authors · 2021-08-09
> **Answer to reviewer pxsk**
>
> We would like to thank the reviewer for their insightful feedback and interesting observations. We are encouraged that the reviewer found the work novel, the experiments and ablations thorough, the claims backed up, and the paper well written and easy to follow. We thank the reviewer for their support of our work. We address the reviewer’s comments below and will include all the feedback in the revised version of the manuscript.
>
> “**I think adding some metric of diversity in addition to FID/IS to at least some experiments would help**.”
> It is indeed beneficial to report additional metrics emphasizing diversity. We will therefore additionally include Recall as a measure of diversity, following [1]. We evaluate IC-GAN, class-conditional IC-GAN, class-conditional BigGAN and other baselines for the unlabeled setup when possible on 128x128 and 256x256 ImageNet dataset. We computed Recall across 10,000 generated and real samples and using 5 nearest neighbors, as proposed in [2].
>
> In Table A, we provide Recall values to complement the results already reported in the paper (Figure 4 right). IC-GAN obtains better Recall (and therefore more diversity) than all the baselines in both the unlabeled and labeled settings, further emphasizing what has been shown qualitatively in the paper. Moreover, in line with the plots provided in Figure 4, the diversity of the generated data improves when increasing the number of instances.
>
> **Table A** *Results for ImageNet in terms of Recall(R) [1] (bounded between 0 and 100), using 10,000 real and generated images*.  *"Instance selection", only used for IC-GAN, indicates whether 1,000 conditioning instances are selected with k-means (k-means 1,000)* *or 10,000 conditioning instances are sampled uniformly (random 10,000) from the training set to obtain 10,000 generated images in both cases. \*: Generated images obtained with the paper’s opensourced code*.
>
> | Method      | Res. | Instance selection | R  |
> | ----------- | ----------- |  ----------- | ----------- |
> |*Unlabeled setting*|  |   |   |
> | Self-cond. GAN [3] \* | 128 |  -  |  48.4 $\pm$ 0.8 |
> | IC-GAN | 128 | k-means 1,000 | 55.6 $\pm$ 0.9 |
> | IC-GAN | 128 | random 10,000 |  **69.7** $\pm$ 0.9 |
> | |  |   |  |   |
> | IC-GAN | 256 | k-means 1,000 | 54.3 $\pm$ 0.7 |
> | IC-GAN | 256 | random 10,000 | **68.9** $\pm$ 0.3 |
> |*Labeled setting*|  |   |   |
> | BigGAN [4] | 128 |  -  |  64.2 $\pm$ 0.7 |
> | Class-conditional IC-GAN | 128 | k-means 1,000 | 64.3 $\pm$ 0.7 |
> | Class-conditional IC-GAN | 128 | random 10,000 |  **73.6** $\pm$ 0.5 |
> | |  |   |  |   |
> | BigGAN [4] | 256 |  -  |  70.2 $\pm$ 0.7 |
> | Class-conditional IC-GAN | 256 | k-means 1,000 | 70.4 $\pm$ 0.3 |
> | Class-conditional IC-GAN | 256 | random 10,000 |  **79.3** $\pm$ 0.2 |
>
>
>
>  “**I would have liked a little bit more discussion about what the authors think is being transferred**.” Thanks for the interesting question. In the Supplementary Material, Section F, we will include the following discussion to cover the intuitions on what is being transferred: “From the point of view of KDE, what is being transferred is the kernel shape, not the kernel location (that is controlled by instances). The kernel shape is predicted using a generative model from each input instance and we probe the kernel via sampling from the generator. Thus, we transfer a function that predicts kernel shape from a conditioning, and this function seems to be robust to diverse instances as shown in the paper (e.g. see Fig 1 c-d)).
>
> Moreover, by visually inspecting the generated images in our transfer experiments, we observed that when transferring an IC-GAN trained on ImageNet to COCO-Stuff, if the model is conditioned on images that contain unseen classes in ImageNet, such as “giraffe”, the model will still generate an animal that would look like a giraffe without the skin patterns and characteristic antennae, because ImageNet contains other animals to draw inspiration from. This suggests that the model generates plausible images that have some similar features to those present in the instance conditioning, but adapting it to the training dataset style. Along these lines, we also observed that in some cases, shapes and other object characteristics from one dataset are transferred to another (ImageNet to COCO-Stuff). Moreover, when we conditioned on instances from Cityscapes, the generated images were rather colorful, resembling more the color palette of ImageNet images rather than the Cityscapes one.”
>
>
> [1] Sajjadi, M. S., Bachem, O., Lucic, M., Bousquet, O., & Gelly, S. (2018). Assessing generative models via precision and recall. arXiv preprint arXiv:1806.00035.
>
> [2] Naeem, M. F., Oh, S. J., Uh, Y., Choi, Y., & Yoo, J. (2020, November). Reliable fidelity and diversity metrics for generative models. In International Conference on Machine Learning (pp. 7176-7185). PMLR.
>
> [3] Liu, S., Wang, T., Bau, D., Zhu, J. Y., & Torralba, A. (2020). Diverse image generation via self-conditioned gans. In Proceedings of the IEEE/CVF Conference on Computer Vision and Pattern Recognition (pp. 14286-14295).
>
> [4] Brock, A., Donahue, J., & Simonyan, K. (2018). Large scale GAN training for high fidelity natural image synthesis. arXiv preprint arXiv:1809.11096.

---

> > ### Comment · Reviewer_pxsk · 2021-08-27
> > **Response**
> >
> > Thank you!
> >
> > I think the transfer learning stuff sounds really interesting. I maintain my rating.

---

### Decision · Program_Chairs · 2021-09-27

**Decision:**

Accept (Spotlight)

**Comment:**

Two of the reviewers strongly support the paper and find the idea original. Another reviewer thought that the ability to perform good transfer learning is also of interest to the community. I agree with their observations and recommend acceptance.